# Learning Successor Features with Distributed Hebbian Temporal Memory

**Evgenii Dzhivelikian**
MIPT, AIRI
Moscow, Russia
dzhivelikian.ea@phystech.edu

**Petr Kuderov**
AIRI, MIPT
Moscow, Russia
kuderov@airi.net

**Aleksandr Panov**
AIRI, MIPT
Moscow, Russia
panov@airi.net

## Abstract

This paper presents a novel approach to address the challenge of online sequence learning for decision making under uncertainty in non-stationary, partially observable environments. The proposed algorithm, Distributed Hebbian Temporal Memory (DHTM), is based on the factor graph formalism and a multi-component neuron model. DHTM aims to capture sequential data relationships and make cumulative predictions about future observations, forming Successor Features (SFs). Inspired by neurophysiological models of the neocortex, the algorithm uses distributed representations, sparse transition matrices, and local Hebbian-like learning rules to overcome the instability and slow learning of traditional temporal memory algorithms such as RNN and HMM. Experimental results show that DHTM outperforms LSTM, RWKV and a biologically inspired HMM-like algorithm, CSCG, on non-stationary data sets. Our results suggest that DHTM is a promising approach to address the challenges of online sequence learning and planning in dynamic environments.

## 1 Introduction

Modelling sequential data is one of the essential tasks in artificial intelligence as it has many applications, including decision-making and learning world models (Moerland et al., 2023), natural language processing (Min et al., 2021), conversational AI (Dwivedi et al., 2023), time series analysis (Eraslan et al., 2019), and video and music generation (Ji et al., 2020). One of the classical approaches to modelling sequential data is to form a representation that stores and condenses the most relevant information about a sequence, and to find a general transformation rule of this information through the dimension of time (Lipton et al., 2015; Harshvardhan et al., 2020; Mathys et al., 2011). We refer to the class of algorithms that use this approach as temporal memory (TM) algorithms because they essentially model the cognitive ability of complex living organisms to remember the past experience and make future predictions based on that memory (Hochreiter & Schmidhuber, 1997; Friston et al., 2016; 2018; Parr & Friston, 2017).

Temporal memories differ in their ability to capture data dependencies and the extent to which they can generalise their predictions to unseen sequences. There are two extremes known in psychology as cognitive maps Tolman (1948), which extract commonalities or schemas generalising experience, and episodic memory Tulving et al. (1972), which stores fragments of experience separately. They both can be viewed as temporal memories, however, the former is slower and require a lot of data to built up a useful schema, according to contemporary computational models based on HMM (George et al., 2021) and Transformers (Dedieu et al., 2024), while the latter is instant and is usually viewed as a data buffer (Zhang et al., 2021).

In this paper, we explore advantages of an episodic-like memory in addressing the problem of online sequence learning for planning and decision-making under uncertainty, which can be formalized as reinforcement learning (RL) for a partially observable Markov decision process (POMDP) (Poupart, 2005; Singh et al., 2021). Inferring the hidden state in a partially observable environment is a sequence modelling problem, since it requires processing a sequence of observations to obtain enough information about the hidden state. One of the most efficient representations of hidden states for discrete POMDPs is the Successor Representation (SR), which disentangles hidden states and goals

given by the reward function (Dayan, 1993; Samuel J. Gershman, 2018; Millidge & Buckley, 2022). The Successor Features framework is an extension of SR to continuous POMDP, using the same idea of value function decomposition, but instead for features of a hidden state (Barreto et al., 2017; Touati et al., 2022). Temporal memory (TM) algorithms can be used to make cumulative predictions about future states and their features, forming SR or SF. This work shows that the proposed algorithm, Distributed Hebbian Temporal Memory (DHTM), can dynamically form SFs for navigation tasks in Gridworld and AnimalAI (Fountas et al., 2020) environments.

The most prominent TM algorithms, such as a Recurrent Neural Network (RNN) (Boden, 2002) or LSTM (Hochreiter & Schmidhuber, 1997), use backpropagation to capture data relationships, which is known to be unstable due to recurrent nonlinear derivatives. They also require complete data sequences to be available during training. Although the gradient vanishing problem can be partially circumvented in a way that Receptance Weighted Key Value (RWKV) (Peng et al., 2023) or Linear Recurrent Unit (LRU) (Orvieto et al., 2023) models do, the problem of online learning, i.e. learning on non-stationary data sets, is still a viable issue. Unlike graphical models such as HMM (Eddy, 2004), RNN models and their descendants also lack a probabilistic theoretical foundation, which is advantageous for modelling sequences captured from stochastic environments (Salaün et al., 2019; Zhao et al., 2020). There is little research on TM models that can be used in fully online adaptive systems interacting with partially observable stochastic environments with access to only one sequence data point at a time, a common case in reinforcement learning (Jahromi et al., 2022).

We propose a Distributed Hebbian Temporal Memory (DHTM) algorithm based on factor graph formalism and a multi-compartment neuron model inspired by the HTM neuron (Hawkins & Ahmad, 2016). Factor graph is a flexible graphical representation of probabilistic distributions, which is commonly used in combination with sum-product algorithm (Loeliger, 2004). We found this representation particularly convenient in our attempt of probabilistic interpretation of the HTM framework. The resulting graphical structure of our model is similar to that of the factorial HMM (Ghahramani & Jordan, 1995). An important feature of our model is that the transition matrices for each factor are stored as different components (segments) of artificial neurons, which makes computation very efficient in the case of sparse transition matrices. Our TM forms sequence representations entirely online and uses only local Hebbian-like learning rules (Hebb, 2005; Churchland & Sejnowski, 1992; Lillicrap et al., 2020), which avoid the drawbacks of gradient methods and make the learning process much more sample efficient than gradient methods. This comes at the expense of hidden state generalization, since DHTM's hidden state uniquely encodes each observation sequence. Different trajectories leading to the same position in the partially observable Gridworld can result in different hidden states inferred by DHTM. In essence, this makes DHTM a model of episodic memory, or an adaptive trajectory buffer, with built-in machinery for making multistep future predictions. In this light, we also show that an agent forming SF with DHTM can be interpreted as a kind of episodic control (EC) (Lengyel & Dayan, 2007; Pritzel et al., 2017; Ritter et al., 2018; Emukpere et al., 2023).

Though our approach is similar to EC, we are not aware of other works, demonstrating its application to partially observable environments. There are other temporal memories that, similarly to DHTM, use Hebbian rules and learn online, like Asymmetric Hopfield Network (AHN) (Chaudhry et al., 2023), Temporal Predictive Coding (tPC) (Tang et al., 2023) and Hierarchical Temporal Memory (Hawkins & Ahmad, 2016). Compared to DHTM, AHN is a first-order memory. That is, it is not able to disentangle identical observations that correspond to different underlying environmental states. Therefore, it is not suitable for partially observable environments. In contrast, tPC has a hidden layer that, potentially, makes it a high-order memory. Despite that it has been tested only in MDP environments (Tang et al., 2024) so far, this method paves a promising direction, which we aim to compare with in future work. Finally, the algorithm that inspired our work, HTM, is also a Hebbian high-order temporal memory, which, opposed to DHTM, doesn't allow for probabilistic predictions due to its all-or-nothing computations.

The proposed TM is tested as an episodic memory for an RL agent architecture navigating in a Gridworld environment and a more challenging AnimalAI testbed (Crosby et al., 2020). The results demonstrate that our algorithm outperforms classical LSTM, RWKV and a biologically inspired HMM-like world model CSCG (George et al., 2021) in changing environments and is in order of magnitude more sample efficient than Dreamer V3 (Hafner et al., 2023). Another advantage of our algorithm is that it can be implemented on neuromorphic processors using only local learning rules.

Our contribution to this work includes the following key elements:

- A novel episodic memory model, called DHTM, which is based on a factor graph formalism and features an efficient, bio-inspired implementation.

- We demonstrate that the DHTM can operate fully online, using only local Hebbian-like learning rules.

- We show that DHTM learns at a speed comparable to model-based table episodic control while supporting distributed representations. This capability significantly broadens its applicability.

## 2 BACKGROUND

### 2.1 REINFORCEMENT LEARNING

This paper considers decision-making in a partially observable environment (Poupart, 2005). The environment is defined by its state transition function $P(s, a, s') = Pr(s' \mid s, a)$, determining its dynamics, where $s', s \in S$ is called state space, $a \in A$—action space. Partial observability means that an agent has access to observations $o \in O$, which only partially inform about the environment's state $s$ at each moment.

In reinforcement learning (RL), the goal for the agent is set through the reward function $R(s) : S \to \mathbb{R}$. The task of RL is to find the agent's policy $\pi(a \mid s) : S \times A \to [0, 1]$ that maximizes the expected return $G = \mathbb{E}[\sum_{t=0}^{T} \gamma^l R_t]$, where $T$ is an episode length and discount factor $\gamma$ sets the importance of remote rewards. Value-based methods usually aim to estimate the Q-function given a policy $\pi$: $Q^\pi(s_t, a_t) = \mathbb{E}[\sum_{l \geq t} \gamma^{l-t} R(s_{l+1}) \mid s_t, a_t, \pi]$. For an optimal value function $Q^*$, an optimal policy can be defined as $\pi(a \mid s) = \operatorname*{argmax}_a Q^*(s, a)$.

One of the drawbacks of value-based methods is that they require learning a new Q-function when the reward function changes, even if the environment remains the same. To get around this, the dynamics of the environment and the reward function can be decoupled, which leads us to the successor representation framework.

### 2.2 SUCCESSOR REPRESENTATION

Successor representations are such representations of hidden states from which we can linearly infer the state value given the reward function (Dayan, 1993). For a discrete state space, such representation can be derived just by rearranging sums in the equation for value function:

$$V(h_t = i) = \mathrm{E}[\sum_{l=0}^{\infty} \gamma^l R_{t+l+1} \mid h_t = i] = \sum_j \mathrm{SR}_{ij} R_j, \tag{1}$$

where $\gamma$ is a discount factor, $i$-th row in the matrix SR is the Successor Representation of a state $i$, and $\mathrm{SR}_{ij} = \sum_{l=0}^{\infty} \gamma^l p(h_{t+l+1} = j \mid h_t = i)$. $R_j$ is a reward for observing the state $j$.

An extension of SR for large or continuous state spaces that we use is Successor Features (Barreto et al., 2017), which is based on a similar idea of value function decomposition, but in a feature space instead (see details in Appendix A).

It is easy to show that, similarly to the Q-function, both SR and SF can be obtained using dynamic programming. However, this limits its use to relatively stable environments, since changes in the transition function can't be taken into account immediately.

The next step to improve an agent's adaptability is to use a world model to approximate the environment's transition function and use it directly to form SR according to its definition. And one of the most basic such world models is the Hidden Markov Model.

### 2.3 HIDDEN MARKOV MODEL

A partially observable environment can be approximated by a Hidden Markov Model (HMM) with hidden state space $H$ and observation space $O$. Generally, $H$ is not equal to environemnts state

space $S$ since the environemnt's true structure is usually unknown. For simplicity, we assume that actions are fully observable and that information about them is contained in hidden state. For the process of length $T$ with state values $h_{1:T} = (h_1, \ldots, h_T)$ and observations $o_{1:T} = (o_1, \ldots, o_T)$, the Markov property yields the following factorization of the generative model:

$$p(o_{1:T}, h_{1:T}) = p(h_1) \prod_{t=2}^{T} p(h_t|h_{t-1}) \prod_{t=1}^{T} p(o_t|h_t). \tag{2}$$

In the case of a discrete hidden state variable, a time-independent stochastic transition matrix can be learned with the Baum-Welch algorithm (Baum et al., 1970), a variant of the expectation maximization (EM) algorithm. To compute the statistics for the expectation step, it uses the forward-backward algorithm, which is a special case of the sum-product algorithm (Kschischang et al., 2001).

Factor graph is a convenient graphical representation (Loeliger, 2004) of probabilistic models like HMM (see Fig.1A for an example). Formally, a factor graph is a bipartite graph, where one part consists of factors (squares), which, replacing conditional probabilities, define dependencies between random variables (circles) constituting the other part. Such representation also allows extension of forward-backward algorithm to sum-product algorithm. As shown in section 3.1, we base DHTM on ideas from the sum-product algorithm and the Factorial HMM, while adapting them for online learning.

## 3 DISTRIBUTED HEBBIAN TEMPORAL MEMORY

### 3.1 FACTOR GRAPH MODEL

Distributed Hebbian Temporal Memory is based on the sum-product belief propagation algorithm in a factor graph (see Figure 1A). Analogous to Factorial HMM (Ghahramani & Jordan, 1997), we divide the hidden space $H$ into subspaces $H^k$. There are four sets of random variables (RV) in the model: $H_{t-1}^i$—latent variables representing hidden states from the previous time step (context), $H_t^k$—latent variables for the current time step, and $\Phi_t^k$–feature variables. All random variables have categorical distributions. RV state values are denoted by corresponding lowercase letters: $h_{t-1}^i$, $h_t^k$, $\varphi_t^k$, $o_t^{lm}$.

In this work, each variable $\Phi_t^k$ is considered independent and has a separate graphical model for increased computational efficiency. However, in practice, hidden variables of the same time step are statistically interdependent. We introduce their interdependence through a segment computation trick that goes beyond the standard sum-product algorithm (see Appendix B for details).

The model also has three types of factors: $M_{t-1}^i$—messages from previous time steps, $F_c^k$—context factor (generalized transition matrix), $F_e^k$–emission factor. We assume that messages $M_{t-1}^i$ contain posterior information from time step $t-1$, so we don't plot observable feature variables for previous time steps in Figure 1A.

We will discuss only the upper block of the graph in Figure 1A, which is DHTM itself. The lower block—an encoder—is described in Appendix D. The only requirement for the encoder is that its output should be represented as states of categorical variables (features) for the current observation.

The main routine of the DHTM is to estimate distributions of hidden state variables given by the equation 3, the computational flow of which is schematically shown in Figure 1B:

$$p(h_t^k) \propto \sum_{\Omega_k} \prod_{i \in \omega_k} M_{t-1}^i(h_{t-1}^i) F_c^k(h_t^k, \Omega_k), \tag{3}$$

where summation is over all possible hidden RV states in $\Omega_k = \{h_{t-1}^i : i \in \omega_k\}$, $\omega_k = i_1, \ldots, i_n$—set of previous time step RV indexes included in $F_c^k$ factor, $(n+1)$—factor size.

### 3.2 NEURAL IMPLEMENTATION

For computational purposes, we translate the factor graph shown in Figure 1A to the Hebbian learning neural network architecture shown in Figure 1B (for a biological interpretation of the model,

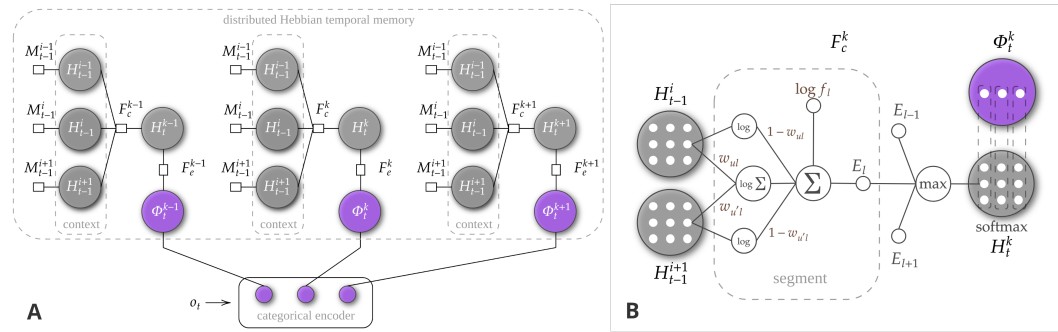

Figure 1: **A.** Example of a factor graph for the DHTM. The input to the model is a sequence of observations $o_t$. The encoder block forms categorical features $\Phi_t^k$. Each feature $\Phi$ has its own explaining hidden variable, which may depend on hidden variables of the other features and on itself from the previous time step. $F_c^k$ and $F_e^k$ are context and emission factors for the corresponding variables. Unary factors $M_{t-1}^i$ called messages represent accumulated information about previous time steps. **B.** Neural implementation. Random variables are represented by cell clusters (white circles), where each cell corresponds to a state and its spike frequency to the probability of the state $p(h_t^k)$. The dendritic segments of the cell correspond to the context factor values $f_l$ for a given combination of states (active presynaptic cells). Segments' excitations $E_l$ are combined to determine the cell's spike frequency $p(h_t^k)$. Segment synaptic weights reflect the specificity of the combination of presynaptic cells for the segment. Emission factors $F_e^k$ are fixed and represented by minicolumns within a variable.

see Appendix C). Each RV can be viewed as a set of neurons representing the states of the RV, i.e. $p(h_t^k) = p(c_t^j = 1)$, where $j$–is the index of a neuron corresponding to the state $h_t^k$ and $c_t^j \in \{0, 1\}$—cell's activity. Since RVs are categorical, only one cell per variable can be active at a time and their activity is defined through sampling of RV's distribution.

Since factors $F_c^k$ represent dependencies between variables, in neural implementation we view them as collections of connections between cell groups corresponding to RVs that $F_c^k$ takes as arguments. Each factor's entry maps a particular combination of states $h_t^k, h_{t-1}^{i_1}, \ldots, h_{t-1}^{i_{n_l}}$ to a factor value $f_l$. We found it useful to represent each such entry as a separate computational unit, which we call segment (alluding to dendritic segments of a biological neuron). As shown in Figure 1B, segment $l$ connects its presynaptic cells $h_{t-1}^{i_1}, \ldots, h_{t-1}^{i_{n_l}}$ (one per each RV), constituting its receptive field, to the cell it excites $h_t^k$. The factor $f_l$ value can be thought of as a segment's excitation efficiency, i.e. how strong its presynaptic pattern influences the cell's activation.

Reformulation of belief propagation step in equation 3 in terms of segmental computations results in the following equations:

$$p(h_t^k) = \operatorname{softmax}_{j \in \text{cells}[H_t^k]} \left( \max_{l \in seg(j)} (E_l) \right), \tag{4}$$

$$E_l = \log f_l + w_l \log \frac{1}{n_l} \sum_{u \in \text{rec}(l)} w_{ul} m_u + \sum_{u \in \text{rec}(l)} (1 - w_{ul}) \log m_u, \tag{5}$$

where $\text{cells}[H_t^k]$—indexes of cells corresponding to states of the variable $H_t^k$, $seg(j)$—all segments for cell $j$, $w_{ul}$—synapse efficacy or neuron's specificity for segment, such that $w_{ul} = p(s_l = 1 | c_{t-1}^u = 1)$, $w_l = \frac{1}{n_l} \sum_u w_{ul}$—average synapse efficacy for segment $l$, and $n_l$-number of cells in segment's receptive field $rec(l)$. $m_u$ is presynaptic cell's activation probability $p(c_{t-1}^u = 1)$, which carries messages from the previous time-step and corresponds to entry in $M_{t-1}^i$. For detailed derivation and rationale behind these equations, see Appendix B.

Initially, cells have no segments. As learning progresses, new non-zero connections, grouped into segments, are created. In equation 5, we benefit from having sparse factors because their complexity depends linearly on the number of non-zero entries or segments. And that's usually the case in our

model due to the one-step Monte Carlo learning and the special form of the emission factors $F_e^k$:

$$F_e^k(h_t^k, \varphi_t^k) = \mathbb{I}[h_t^k \in \mathrm{col}(\varphi_t^k)], \tag{6}$$

where $\mathbb{I}$—indicator function, $\mathrm{col}(\varphi_t^k)$ is a set of hidden states connected to the feature state $\varphi_t^k$ forming a column. The shape of the emission factor is inspired by the presumed columnar structure of the neocortex and has been shown to induce a sparse transition matrix in HMM (George et al., 2021).

The next step after computing the $p(h_t^k)$ distribution is to incorporate information about the current feature states $p(h_t^k \mid \varphi_t^k) \propto p(h_t^k)\mathbb{I}[h_t^k \in \mathrm{col}(\varphi_t^k)]$. After that, the learning step is executed. The loop closing step of our TM algorithm is to assign the posterior of the current step $p(h_t^k \mid \varphi_t^k)$ to $M_{t-1}^i$.

### 3.3 LEARNING

DHTM learns $f_l$ and $w_{ul}$ weights by Monte-Carlo Hebbian-like updates. First, $h_{t-1}^i$ and $h_t^k$ are sampled from their posterior distributions: $p(h_{t-1}^i \mid \varphi_{t-1}^i) \propto M_{t-1}^i$ and $p(h_t^k \mid \varphi_t^k)$ respectively. Cells that correspond to sampled states become active. For cell $j$ corresponding to $h_t^k$ we check if there are segments matching the pattern of sampled states (active cells) from the previous time-step $\{h_{t-1}^i\}_i$. If there are no such segments, the new segment connecting sampled cells is created. For all newly created segments and segments with receptive fields matching active cells from the previous time-step, its activity $s_l$ is set to 1. Then $f_l$ is updated according to the activity of the segment $s_l$ and the activity of its cell $c_t^j$, so that $f_l$ is proportional to several coincidences $s_l = c_t^j = 1$ in the recent past (exponential moving average of coincidence):

$$\Delta f_l = \alpha(c_t^j - f_l)s_l, \tag{7}$$

where $\alpha \in [0, 1)$ is the segment's learning rate.

That is, $f_l$ is trained to be equal to the cell's average activation frequency when the segment is active. It's similar to Baum-Welch's update rule (Baum et al., 1970) for the transition matrix in HMM, which, in effect, counts transitions from one state to another, but in our case, the previous state (context) is represented by a group of RVs.

Weights $w_{ul}$ are also updated by the Hebbian rule to reflect the specificity of a presynaptic $u$ for activating a segment $l$. That is, they are targeted to represent the probability $p(s_l = 1 \mid c_{t-1}^u = 1)$ that segment $s_l$ is active, given that cell $u$ was active at the previous time step. In our algorithm, it is approximated as an exponential moving average of the frequency activation of segment $s_l$, given $c_{t-1}^u = 1$:

$$\Delta w_{ul} = \beta \cdot \mathbb{I}[c_{t-1}^u = 1] \cdot (\mathbb{I}[s_l = 1] - w_{ul}), \tag{8}$$

where $\beta \in [0, 1)$ — learning rate.

### 3.4 AGENT ARCHITECTURE

We incorporate DHTM as a part of an RL agent. The same agent is used for other memories tested. The agent consists of a memory model and a feature reward function. The memory model generates SF by predicting cumulative future distributions of feature variables $\Phi^k$. We assume that the reward function can be decomposed linearly in feature space as $R_t = \frac{1}{n} \sum_{k=1}^n r_i^k$, where $r_i^k$ is a reward associated with feature state $\varphi_t^k = i$ and $n$—number of feature variables. $r_i^k$ is learned during interaction with the environment and is used in combination with SF representations to estimate the action value function (see Appendix A for details).

The agent training procedure is outlined in Algorithm 1. For each episode, the memory state is reset to a fixed initial message with RESET_MEMORY() and the variable `action` is set to a fixed initial action. An observation returned by an environment (`obs`) is encoded as a set of categorical variables. In the OBSERVE() routine, memory learns to predict next feature states as described in section 3. An agent learns associations to map feature states and rewards in the REINFORCE() function:

$$r_i^k \leftarrow r_i^k + \alpha\mathbb{I}[\varphi_t^k = i](R_t - r_i^k) \tag{9}$$

where $\alpha$ is a learning rate, $R_t$—a reward for the current time step.

**Algorithm 1** General agent training procedure

```
 1: for episode=1..n do
 2:     RESET_MEMORY()
 3:     action ← initial_action
 4:     while (not terminal) and (steps < max_steps) do
 5:         obs, reward ← STEP()
 6:         features ← ENCODE(PREPROCESS(obs))
 7:         OBSERVE(features, action)
 8:         REINFORCE(reward, features)
 9:         action ← SAMPLE_ACTION()
10:         ACT(action)
11:     end while
12: end for
```

We include actions in DHTM by forcing some of the hidden variables $H_t^k$ to represent actions. That is, we assume that information about action is contained in the hidden state of the model. For example, if we have 4 actions, we set 4 states for one of the hidden variables and set its state from observing the action.

In SAMPLE_ACION(), for each action and current state, TM predicts feature states up to time step $T$ under policy $\pi$, and the predictions are combined to form SF (equation 10). For simplicity, we always build SF under a uniform policy, which doesn't guarantee optimality, but we found that it doesn't make a significant difference to iterative policy improvement in our experimental setups. The planning horizon $T$ is determined dynamically from the distribution of $\Phi_{t+l}^k$. For each prediction step $l$, we check whether the distribution of $\Phi_{t+l}^k$ is close to uniform (according to Kullback-Leibler distance) or whether any of states with positive reward has probability above a threshold; in that case, the prediction cycle is stopped. The action value is determined by combining the corresponding SF and feature state rewards as shown in equation 11. Finally, the action is sampled from the softmax distribution over the action values.

$$\text{SF}_{t+T}^\pi(\varphi^k = j \mid h_t) = \sum_{l=0}^{T} \gamma^l p_\pi(\varphi_{t+l+1}^k = j \mid h_t) \tag{10}$$

$$Q^\pi(h_t, a_t) = \sum_{jk} \sum_{h_{t+1}} \text{SF}_{t+T}^\pi(\varphi^k = j \mid h_{t+1}) \cdot p(h_{t+1} \mid h_t, a_t) \cdot r_j^k, \tag{11}$$

## 3.5 DHTM and Episodic Control

**Algorithm 2** Episodic memory learning

**Input:** $\varphi_{t+1}, a_t$
```
 1: h_{t+1} ← D_{a_t}(h_t)
 2: φ*_{t+1} ← argmax F_e(h_{t+1}, φ)
              φ
 3: if h_{t+1} is null or φ*_{t+1} is not φ_{t+1} then
 4:     h_{t+1} ← argmax F_e(h, φ_{t+1}) # K is the
              h∉K
        set of keys for all actions
 5:     D_{a_t}(h_t) ← h_{t+1}
 6: end if
 7: h_t ← h_{t+1}
```

It can be shown that DHTM combined with the agent architecture described in the previous section is similar to episodic control.

DTHM performs best when its hidden state space is large. Since the hidden state for unpredicted observations is formed randomly, it's important that there are no accidental state collisions when remembering transitions. That is, DHTM tries to remember new sequences without interfering with old memories. In effect, this makes DHTM a buffer of agent trajectories with a built-in recall engine. Episodic Control is based on a similar idea, and to investigate its difference from DHTM, we implemented an Episodic Control agent, a simplified counterpart of DHTM, reflecting its core property that presumably helps it solve RL tasks.

Episodic Control (EC) is a general computational approach to storing and reusing an agent's experience, inspired by the notion of episodic memory from psychology (Tulving et al., 1972). The core of EC is a dictionary $\mathcal{D}_a = (K_a, V_a)$ that stores key-value pairs for each discrete action $a \in A$. In our EC version, $K_a$ is an array of hidden states $h_t$ and $V_a$ is an array of corresponding subsequent hidden states $h_{t+1}$. If the key state $h_t$ already exists in the dictionary, its value is rewritten. That is, $\mathcal{D}_a$ represents the transition matrix of a deterministic HMM for a feature variable $\Phi_t$. The hidden state $h_t$ and the feature state $\varphi_t$ have the same dependence as in DHTM (equation 6), but in EC we assume that there are infinite hidden states per feature state. To predict the next state, we just have to look up $\mathcal{D}_a$ for the current state $h_t$, but if there is no entry for $h_t$ or the prediction does not match the observed state $\varphi_{t+1}$, then the new state is formed. Unlike DHTM, the new state is not completely

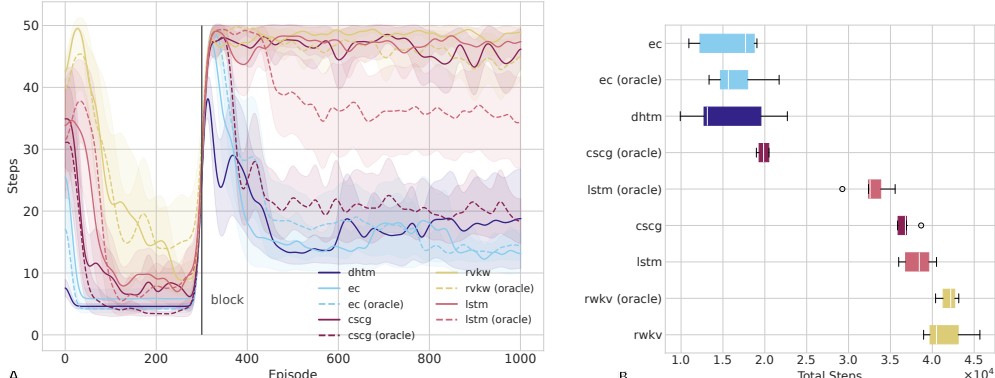

Figure 2: POMDP 5x5 Gridworld foraging task using different types of memory: DHTM, CSCG, LSTM, RWKV. All agents, except for DHTM, are trained in two modes. The first mode is with oracle, which tells the agent to reinitialize memory and clear the trajectory buffer during environmental change. The second mode is without oracle, so the agent continues using its current strategy to gather new trajectories and update memory. There is also an episodic control agent (`ec`) among baselines, which uses a simple dictionary to store trajectories and make predictions. **A.** Dynamic of steps required for the agent to reach the goal. In the 300th episode, the goal is blocked by a wall, so the agent needs to find a new optimal trajectory. **B.** Total steps taken during 1000 episodes.

random, but to avoid collisions, the algorithm makes sure that this state hasn't been chosen before. The learning procedure of the EC agent's memory $\mathcal{D}_a$ is summarized in the algorithm 2.

Since $\mathcal{D}_a$ can be transformed into a transition matrix, the process of forming SF can be reduced to one in Sec. 3.4. However, since the resulting transition matrix is sparse, it's more efficient to consider $\mathcal{D}_a$ as a graph representation and to perform SF formation as a breadth-first search (see Appendix E.3 for details).

## 4 EXPERIMENTS

We test our model in a reinforcement learning task in the Gridworld environment and in a more challenging AnimalAI 3D environment. This section shows how different memory models affect the adaptability of an RL agent. For each setup, the results are averaged over five experiments with different seed values, and one standard deviation is shown everywhere, except for boxplots. See Appendix E for details on the baselines, their model parameters, and training regimes. See Appendix F for a more detailed description of the environments and setups[1].

### 4.1 GRIDWORLD

The first test was conducted in partially observable Gridworld in a 5x5 alternating maze with a stationary end state and the agent's starting position. The agent can observe only the colour of the cell it is in and the colour of an obstacle it encounters, while being punished in addition to the constant step penalty. Each episode ends when the reward is collected or when the maximum number of action steps is reached. In each episode, we measure the number of action steps required for the agent to reach the goal.

Each trial lasts 1000 episodes and consists of two phases. In the first phase, there are no obstacles on the shortest path to the terminal state. In the second phase, after 300 episodes, the previously optimal path is blocked by a wall, so the agent has to take a detour to get the reward. The experiment aims to show the ability of a memory to relearn the structure of the maze in the test time. If the agent is able to reach the end state in the second phase, it suggests that the model can forget, or at least suppress,

---

[1]The source code of the experiments and algorithms is at `https://github.com/Cognitive-AI-Systems/him-agent/tree/iclr25`.

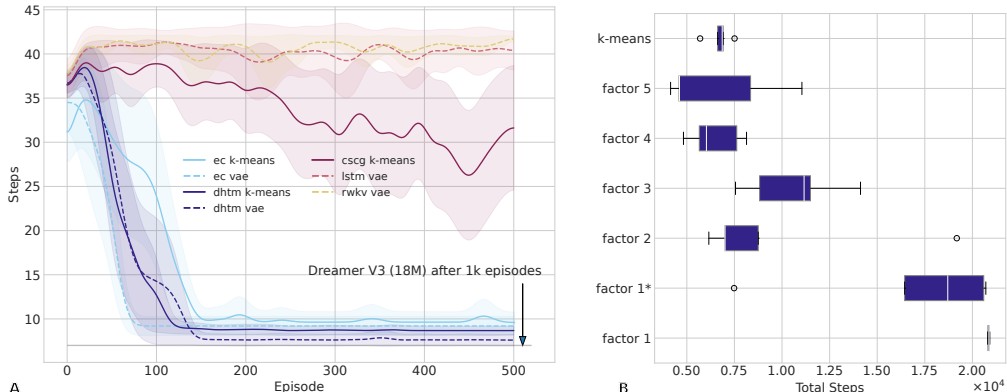

Figure 3: Amount of action steps the agent performs to reach the goal in a 10x10-meter room in AnimalAI environment using DHTM, CSCG, LSTM, RWKV and EC dictionary. The agent and the goal have the same initial positions throughout the episodes. Each time-step, the agent gets an encoded image of a first-person view of the environment and produces one of three actions: turn left, turn right, or go forward. **A.** Dynamic of steps required for the agent to reach the goal. **B.** Total steps taken during 1000 episodes by the agent with DHTM with K-Means encoder and Categorical VAE. For the latter, DHTM's context factor size is varied (i.e., how many previous time-step hidden variables are considered for prediction).

the memory of the old shortest path when it is blocked. Otherwise, the agent will continue to hit the wall and fail to reach the terminal state.

In this experiment, we compare our memory model DHTM with CSCG (George et al., 2021), LSTM (Hochreiter & Schmidhuber, 1997) and RWKV (Peng et al., 2023) within the same agent architecture that uses memory to form successor features and evaluate actions (see Sec. 3.4). All memories, except DHTM, are trained in two modes. The first mode is with an oracle that tells the agent to reinitialize memory and clear the trajectory buffer just in time for the environmental change. The second mode is without an oracle, so the agent continues to use its current strategy to collect new trajectories and update memory in the second phase. There is also a simple episodic control agent (EC) among the baselines, which uses a dictionary to store trajectories and make predictions (see Section 3.5 for details).

The results in Figure 2 show that CSCG effectively can't adapt to the environmental change in the online regime, but only if it is retrained from scratch on new trajectories. And, apparently, LSTM and RWKV are only able to learn a predictive model for a small horizon, since they perform significantly worse than other baselines in the second phase, even with oracle enabled. In contrast, DHTM and EC are able to adapt online even without explicit memory's trainable parameters reinitialisation. Another important observation is that the EC agent performs on par with DHTM, suggesting that its mechanism can explain most of DHTM's performance in this task. However, see Appendix H.2 for experiments that show their difference in scalability.

## 4.2 ANIMALAI

AnimalAI is a 3D environment capable of providing rich visual input to an agent in the form of RGB images. The goal of the experiment described here is to show that DHTM can handle distributed representations and is scalable to environments with larger state spaces.

The agent's task is to reach a sphere of radius $0.5\,\mathrm{m}$, which is located in a square area of size 10 by 10 meters. The start position of the agent and the position of the ball are the same between episodes. In each episode, we measure the number of actions (steps), which are AnimalAI's standard left/right turn and forward walk, the agent takes to reach the goal, with the maximum number of actions limited to $40$. Each action is repeated three times and three frames are skipped, so one agent action step corresponds to three steps in the environment.

We test DHTM and EC agents using two different encoders: K-Means and Categorical VAE (Jang et al., 2017). Both encoders are pre-trained on RGB images collected with a uniform strategy during 2000 episodes in the same room used for agent testing. K-Means represents an image as a single categorical variable with 50 states, and Categorical VAE is configured for distributed representation with five variables with 50 states each (see Appendix D for details). In case of EC agent, since it doesn't support distributed representations, we map each unique VAE encoding to one feature state using a simple dictionary. Since the resulting number of states for such feature variable is large (typically more than 1000), it's infeasible to use the same mapping for CSCG. We also didn't find much difference for LSTM and RWKV using K-Means or VAE, so we only keep the results for VAE encoder.

The results are shown in Figure 3A. It can be seen that after about 150 episodes, the agent with DHTM and the EC agent reach the goal stably, although the DHTM agent with distributed encoding (with a factor size of 4) converges to a slightly shorter path on average. The agent with CSCG slowly improves its strategy, but LSTM and RWKV agents don't show any learning on this time-step scale. To understand a common learning time-step scale for contemporary model-based algorithms, we also tested a small version of Dreamer V3 (18M parameters) (Hafner et al., 2023), which successfully solves the same task after about $35 \times 10^3$ action steps (1000 episodes) versus $3.5 \times 10^3$ steps (150 episodes) for DHTM (see Appendix H.1 for details).

To show that DHTM can't simply be replaced by multiple parallel EC dictionaries or CSCGs in the case of distributed encoding, we also vary the context factor size of DHTM, that is, how many previous time-step variables are taken into account to predict the current state of a variable. The factor size of 1 corresponds to the graph of the Factorial HMM, with parallel HMMs. It can be seen from Figure 3B that in our case of categorical VAE, knowing only the previous state of the variable is not enough to reliably predict rewarding features, since the agent doesn't converge to the shortest path. We also tested the variant (`factor 1*`) where each feature variable is represented by three hidden states, to ensure that the result for factor size 1 is not due to DHTM's hidden state collisions. DHTM with a factor that takes into account more than one previous time-step hidden variable (excluding the action variable) performs significantly better. For diagrams depicting factor graphs used in these experiments, see Appendix E.4. For further experiments showing the importance of distributed encoding in DHTM, see Appendix H.3.

## 5  CONCLUSION

This paper introduces a novel episodic memory model, DHTM. It combines ideas from belief propagation, episodic control, and neurophysiological models of the neocortex, resulting in a fast probabilistic sequence memory.

The experiments show that our memory model can quickly learn observation sequences in changing environments and effectively reuse them for RL tasks, on par with model-based table episodic control. We also show that, unlike CSCG and table EC, DHTM is able to handle distributed representations and account for statistical dependencies between variables of these representations. Potentially, it extends the application of our algorithm to higher state spaces, exploiting their factorizations for more efficient computations and to reduce the number of stored parameters.

Our results suggest that DHTM can be viewed as an adaptive trajectory buffer for a slower generalizing algorithm, such as a Neural State Space Model (Jiang et al., 2023). Through prediction error learning, DHTM can selectively discard and replace obsolete transitions in observed trajectories without disrupting relevant experience. It can also provide an initial policy for an agent through episodic control until a slower but more accurate world model takes over. These features of a trajectory buffer are highly desirable in the case of online learning in non-stationary environments.

Further improvements of DHTM can be directed towards the implementation of an automatic feature space factorization discovery to determine an optimal context factor size during learning. To avoid excessive segment growth in noisy environments, it's also possible to implement an adaptive segment activation threshold. One of the main limitations of this study is that the evaluation of the algorithms is limited to basic RL setups due to the simple agent architecture. More advanced use cases could be demonstrated using better exploration strategies and in combination with other memory models, which is also a matter of future research.

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

## A  VALUE FUNCTION DECOMPOSITION

In our agent model, we approximate the reward function R(s) as a sum:

$$R_t = \frac{1}{n} \sum_{k=1}^{n} r(\varphi_t^k), \tag{12}$$

where $r(\varphi_t^k)$ is a reward associated with state $\varphi_t^k$, $n$–number of feature variables. Then, similarly to the Successor Representation idea (see Section 2.2), the value function can be represented as:

$$
\begin{aligned}
V(h_t) = \mathrm{E}[\sum_{l=0}^{\infty} \gamma^l R_{t+l+1} \mid h_t] &= \sum_{l=0}^{\infty} \gamma^l \mathrm{E}[\frac{1}{n} \sum_{k=1}^{n} r(\varphi_t^k) \mid h_t] \\
&= \frac{1}{n} \sum_{k=1}^{n} \sum_{j} \sum_{l=0}^{\infty} \gamma^l p(\varphi_{t+l+1}^k = j \mid h_t) r_j^k \\
&= \frac{1}{n} \sum_{k=1}^{n} \sum_{j} \mathrm{SF}_j^k(h_t) r_j^k,
\end{aligned} \tag{13}
$$

where $\mathrm{SF}_j^k(h_t) = \sum_{l=0}^{\infty} \gamma^l p(\varphi_{t+l+1}^k = j \mid h_t)$, $h_t = (h_t^1, ..., h_t^n)$—hidden state vector of variables $\{H_t^k\}_k$.

## B  DETAILS OF NEURONAL IMPLEMENTATION

As shown in Figure 1B, each RV can be viewed as a set of neurons representing the states of the RV, i.e. $p(h_t^k) = p(c_t^j = 1)$, where $j$–is the index of a neuron corresponding to the state $h_t^k$. Cell activity is binary $c_t^j \in \{0, 1\}$, and the probability can be interpreted as a spike rate. Factors $F_c^k$ and $M_{t-1}^i$ can be represented as vectors, where elements are factor values for all possible combinations of RV states included in the factor. Let's denote the elements of the vectors as $f_l$ and $m_u$ respectively, where $l$ corresponds to a particular combination of state values $h_t^k, h_{t-1}^{i_1}, \ldots, h_{t-1}^{i_{n_l}}$, and $u$ indexes all neurons representing states of RVs from previous time steps.

Inspired by biological neural networks, we group the connections of a neuron into dendritic segments. A segment acts as an independent computational unit that detects a particular input pattern (a contextual state) defined by its receptive field. In our model, a segment connects the factor value $f_l$ and the excitation $E_l$ induced by the segment $l$ to the cell to which it is connected. The segment is active, i.e. $s_l = 1$ if all its presynaptic cells are active, otherwise $s_l = 0$. Computationally, a segment transmits its factor value $f_l$ to the cell to which it is attached if the context matches the corresponding state combination.

We can now rewrite equation 3 as the following:

$$p(h_t^k) \propto \sum_{l \in \mathrm{seg}(j)} L_l f_l^k, \tag{14}$$

where $L_l = \prod_{u \in \mathrm{rec}(l)} m_u$ is segment's likelihood as long as messages are normalized to probability distributions, $\mathrm{seg}(j)$—indexes of segments that are attached to cell $j$, $\mathrm{rec}(l)$—indexes of presynaptic cells that constitute receptive field of a segment with index $l$.

Initially, cells have no segments. As learning progresses, new non-zero connections, grouped into segments, are created. In equation 14, we benefit from having sparse factor value vectors because their complexity depends linearly on the number of non-zero components. And that's usually the case in our model due to the one-step Monte Carlo learning and the special form of the emission factors $F_e^k$:

$$F_e^k(h_t^k, \varphi_t^k) = \mathbb{I}[h_t^k \in \mathrm{col}(\varphi_t^k)], \tag{15}$$

where $\mathbb{I}$—indicator function, $\mathrm{col}(\varphi_t^k)$ is a set of hidden states connected to the feature state $\varphi_t^k$ forming a column. The shape of the emission factor is inspired by the presumed columnar structure

of the neocortex and has been shown to induce a sparse transition matrix in HMM (George et al., 2021).

The segment likelihood $L_l$ resulting from the sum-product algorithm is computed as if the presynaptic cells were independent. However, this isn't usually the case for sparse factors. To account for their interdependence, we substitute the following equation for segment log-likelihood:

$$\log L_l = w_l \log \frac{1}{n_l} \sum_{u \in \text{rec}(l)} w_{ul} m_u + \sum_{u \in \text{rec}(l)} (1 - w_{ul}) \log m_u, \tag{16}$$

where $w_{ul}$—synapse efficacy or neuron specificity for segment, such that $w_{ul} = p(s_l = 1 | c_{t-1}^u = 1)$, $w_l = \frac{1}{n_l} \sum_u w_{ul}$—average synapse efficacy for segment $l$, and $n_l$-number of cells in segment's receptive field $rec(l)$.

The idea behind the formula is to approximate between two extreme cases. The first is $p(s_l = 1 | c_{t-1}^u = 1) \to 1$ for all $u$, which means that all cells in the receptive field are dependent and part of a cluster, i.e. they fire together. In this case, $p(s_l) = m_u$ for all $u$, but we also reduce the prediction variance by averaging across different $u$. In the opposite case, $p(s_l = 1 | c_{t-1}^u = 1) \to 0$ for all $u$, presynaptic cells don't cluster and the segment activation probability is just a product of the activation probability of each cell.

The resulting equation for belief propagation in DHTM is the following:

$$p(h_t^k) = p(c_t^j = 1) = \underset{j \in \text{cells}[H_t^k]}{\text{softmax}} (\underset{l \in seg(j)}{\max} (E_l)), \tag{17}$$

where $E_l = \log f_l + \log L_l$, cells$[H_t^k]$—indexes of cells representing states for the variable $H_t^k$. Here we also approximate the logarithmic sum with the $\max$ operation, inspired by the neurophysiological model of segment aggregation by cells (Stuart & Spruston, 2015).

The next step after computing the $p(h_t^k)$ distribution parameters is to incorporate information about the current feature states $p(h_t^k | \varphi_t^k) \propto p(h_t^k) \mathbb{I}[h_t^k \in \text{col}(\varphi_t^k)]$. After that, the learning step is executed. The loop closing step of our TM algorithm is to assign the posterior of the current step $p(h_t^k | \varphi_t^k)$ to $M_{t-1}^i$.

## C  BIOLOGICAL INTERPRETATION

Neural implementation of the DHTM is inspired by neocortical neural networks (see Fig. 4). Hidden variables $H^k$ may be considered as populations of excitatory pyramidal neurons in cortical layer L2/3 of somatosensory areas, with lateral inhibition modelled as $\text{softmax}$ function. Staiger & Petersen (2021) showed that neurons in this layer are responsible for temporal context formation.

The neuronal activity at timestep $t$ can be thought to carry messages $M_{t-1}^k$. Messages are propagated through synapses of dendritic segments, which correspond to factors $F_c^k$. Dendritic segments of biological neurons are known to be coincidence detectors of their synaptic input (Stuart & Spruston, 2015). That is, each segment can be thought of as a detector of a particular combination of active cells (states in our interpretation). If we associate a segment's parameter with factor value, a collection of dendritic segments represent a mapping from state space to factor values, representing context factor $F_c$.

Our computational model of neuron and its dendrites is based on HTM neuron model (Hawkins & Ahmad, 2016). In this model, each segment has a receptive field and activates when a number of active cells in this receptive field reaches a threshold. The neuron is active if any of its segments are active. We use the same activation rule, but the threshold is fixed to the receptive field size. Our algorithm of segment's growth on demand is also similar to HTM's. In principle, DHTM can be considired as a probabilistic interpretation of HTM, allowing belief propagation.

Similarly to HTM framework, our temporal memory can be interpreted as a model of cortical layers. Feature variables $\Phi_t^k$ may be considered to represent cells of a granular layer (L4), as they are known to be the main hub for sensory excitation for L2/3. L2/3 cells that have common sensory input from the layer L4 are modelled as columns for particular feature states $\text{col}(\varphi_t^k)$ (Mountcastle, 1997).

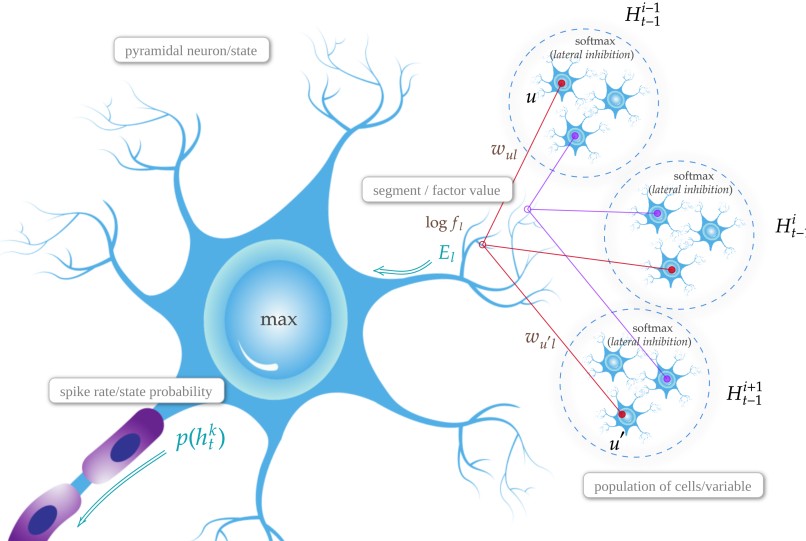

Figure 4: Biological view of the neural implementation of the DHTM. Variables $H_{t-1}$ correspond to populations of neurons that have common sensory input and lateral inhibitory competition. Dendritic segments correspond to factor values $f_l$. Spike frequency of a neuron reflects state probability $p(h_t^k)$ of a variable.

## D  ENCODING OBSERVATIONS

Since DHTM is designed to process categorical random variables, we need a categorical encoder for RGB images to solve AnimalAI tasks.

To compare our model to CSCG and EC in AnimalAI, we use the approach described in Guntupalli et al. (2023) to quantize observation space. K-Means (implementation from `scikit-learn`) with $50$ clusters pretrained on $4 \times 10^4$ 64x64 RGB observations from trajectories sampled by uniform policy. To encode an image, the distance from each cluster is calculated and the index of the closest cluster is returned. In that way, AnimalAI's observation space is represented by one categorical variable.

In order to test DHTM's ability to deal with distributed encoding, we also trained Categorical VAE (Jang et al., 2017) with the latent space represented by five categorical variables with $50$ states each. We use a slightly modified implementation from Subramanian (2020). The only difference is that we modify the loss function by adding mean squared reward prediction error, which is linearly calculated from the latent codes to ensure the linear decomposition in feature space (equation 12). The model is trained on $6 \times 10^4$ 64x64 RGB images for 20 epochs with batch size $64$. An image is encoded by first passing it through VAE's encoder block, then, to ensure stable encoding, instead of sampling from the latent distribution, the state with the highest logit is chosen for each variable.

## E  IMPLEMENTATION DETAILS

As mentioned in Section 3.4, we incorporate DHTM as a part of an RL agent, which has a memory model and a feature reward function. All tested memory models share the same pipeline—conditioned on actions, they learn sequences of categorically encoded observations and are used to form SFs.

In all experiments in Section 4.1, the agent uses softmax exploration with temperature equal to $0.04$, except for EC agent we set softmax temperature to $0.01$, since for higher temperatures its policy is unstable. For SF formation the agent uses $\gamma = 0.8$ and $\gamma = 0.9$ for Gridworld and AnimalAI experiments respectively. Maximum SF horizon is set to $50$, but it is usually shorter since we employ early stop condition based on feature variables distribution. That is, if the prediction is close to uniform or the probability of a state with associated positive reward meets a threshold, the

planning is interrupted. We use probability thresholds equal to $0.05$ and $0.01$ for Gridworld and AnimalAI respectively.

In scaling Gridworld experiments (Appendix H.2), we increase maximum planning horizon up to 100 steps, and we also had to decrease agent's temperature to $0.02$ for DHTM and to $0.002$ for EC agent. Discount factor is $0.7$ for DHTM and $0.9$ for EC. We found that, in 10x10 maze, DHTM's strategy becomes more stable if we increase the positive reward threshold up to $0.4$. EC agent converges faster if we increase its softmax temperature, however, it converges to less optimal trajectory on average.

In the noise tolerance AnimalAI experiment (Appendix H.3), for DHTM we use the same set of agent parameters, except we set the goal probability threshold to $0.5$. We also set the goal detection threshold equal to the mean positive reward since we found that noise variable states mostly get reward assignment slightly above zero, in contrast to signal variables, in which only several states get positive reward assignment.

### E.1 LSTM AND RWKV

LSTM baseline was implemented with a single LSTMCell from PyTorch library (Paszke et al., 2019). It is supported by an additional symexp-layer to encode input before passing it to the LSTM cell and a symexp-layer to decode the LSTM cell's output from the LSTM's hidden state back to the input representation, where symexp activation function, $\text{symexp}(x) = \text{sign}(x)e^{|x|-1}$, is a reverse of symlog function: $\text{symlog} = \text{sign}(x)\log(|x|+1)$.

The similar way we implemented RWKV baseline: a single RWKV layer supported by single-layer encoder and decoder. Current public RWKV implementation is a fast evolving framework (Bo, 2021), and for the increased performance it is tightly bound to the offline batch training common for the transformer architectures. In our case we needed a so-called sequential mode for online learning similar to LSTM. Thus, we adapted another public implementation mentioned in the official documentation (RWKV in 150 lines of code).

For both Gridworld and AnimalAI tasks, LSTM and RWKV with a hidden state size of 100 are trained on a buffer of 1000 trajectories with learning rate $0.002$ on 50 randomly sampled batches of size 50 every 10 episodes.

We also incorporated some notion of random variables in hidden states by splitting the hidden state of the tested RNNs into groups. In all experiments the hidden state represents 10 categorical variables with 10 states. That is, RNN is forced to learn 10 categorical distributions with multi-cross-entropy loss to explain the observed sequences, which is a somewhat close to the multi-categorical hidden state representation used in Dreamer V2/V3 (Hafner et al., 2023). The idea of using symexp activation function, mentioned above, is inspired by Dreamer too, and is used to remedy the problem of learning extreme probability values. Without symexp the neural network has to represent zero probability with high negative logit values and one-probability with high positive logit values, which is hard to reach with low learning rate and may lead to instabilities. Thus, symexp function makes it faster to reach target values in log space.

### E.2 CSCG

CSCG baseline was implemented using code from the repository accompanying the paper (`https://github.com/vicariousinc/naturecomm_cscg`). CSCG was trained on buffers of 500 action steps with 1000 EM steps until convergence. We iteratively calculated the exponential moving average of transition matrices obtained for different batches with a smoothing coefficient $\alpha = 0.05$ as described in the paper (Dedieu et al., 2019). This smoothed transition matrix was used as an initialization for the next training batch and for inference. The first batch of observations is gathered using the uniform policy, and later batches are gathered using the current best policy.

In so-called oracle mode, the transition matrix is trained from scratch every 500 steps. The buffer is discarded at the environmental change and the transition matrix is reinitialised.

### E.3 EPISODIC CONTROL

Here, we provide an algorithm for SF formation using EC dictionary $\mathcal{D}_a$ (see Algorithm 3). In contrast to other memory algorithms, EC dictionary doesn't estimate state transition probabilities, but it stores connections between them in all-or-nothing fation. Therefore, on each time step, all predicted observations are assumed equally probable. But it's not a limitation in our set of experiments, since we tested our algorithms in deterministic environments.

---

**Algorithm 3** SF formation for EC

---

**Input:** $h_0, \gamma \in (0, 1), T$
**Output:** SF
 1: SF $\leftarrow$ array of zeros
 2: PS $\leftarrow \{h_0\}$ # PS is the set of all states (nodes) on the current BFS depth
 3: goal_found $\leftarrow$ false
 4: **for** $l = 1..T$ **do**
 5:     PS $\leftarrow \bigcup_{a \in A} \{\mathcal{D}_a(h)\}_{h \in \mathrm{PS}}$ # get next depth nodes assuming the policy is uniform
 6:     counts $\leftarrow$ array of zeros
 7:     **for all** $h \in$ PS **do**
 8:         $\varphi \leftarrow \underset{\varphi}{\mathrm{argmax}}\, F_e(h, \varphi)$
 9:         counts$_\varphi$ = counts$_\varphi$ + 1
10:         goal_found $\leftarrow r_\varphi > 0$ # check if the feature state is rewarding
11:     **end for**
12:     SF = SF + $\gamma^{l-1}$ NORMALIZE(counts)
13:     **if** goal_found is true **then**
14:         break
15:     **end if**
16: **end for**

---

### E.4 DHTM

In the Gridworld task and AnimalAI with K-Means encoder, the factor-graph for DHTM consists of three hidden variables $H$ with $40$ states per one feature state. Each hidden variable gets the same copied observation, and their predictions are averaged. Using several hidden variables here is essential, as it significantly increases the amount of trajectories memory is able to store by decreasing collision probability for hidden state representations of different trajectories.

In AnimalAI experiments, hidden variables state spaces are of size $40$ states per one feature state. For Categorical VAE encoder, we use one hidden variable per feature variable, except the case (`factor 1*`) that tests that information from one feature variable is not sufficient for correct prediction, in which three hidden variables per observation variable are used and the factor of size three connecting to only hidden copies of the same feature variable. Examples of typical DHTM's graphical models used in our experiments are presented on Figure 5. It's important to note that edge colours denote separate graphical models, but the variables are shared (copied) between them. So there are no loops in these models. For simplicity, we define context factor size as the number of hidden variables from the previous time-step it has connections to, but it also connects to one hidden variable of the current step and the action variable.

The most important DHTM's parameters, except the factor size, are factor learning rate $\alpha$ and its initial value $f_0$. For better adaptability in changing and stochastic environments, the initial factor value should be set closer to zero, but in stationary environments, it can be safely set $f_0 = 1$, allowing faster convergence. In Gridworld we use $\alpha = 0.1$ and $f_0 = 0.05$, and in AnimalAI we set $\alpha = 0.001$ and $f_0 = 1$. We also set the limit on the number of segments $5 \times 10^4$, but practically never reach it in our experiments.

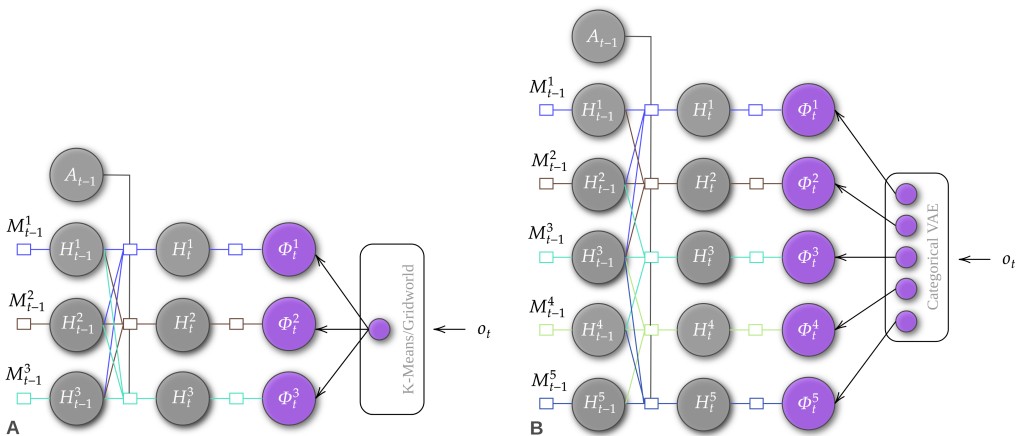

Figure 5: Examples of factor-graphs for two main setups. Edge colours show that edges correspond to separate graphical models with copied variables. All context factors are also connected to action variable $A_{t-1}$. **A.** Graphical model for one feature variable. In that case, feature value is copied for three hidden variables, to increase hidden state space and avoid memory collisions. **B.** Graphical model for Categorical VAE distributed encoding, example for factor of size 3 (i.e. takes into account tree hidden variables from the previous time-step).

## F EXPERIMENTAL SETUPS

### F.1 GRIDWORLD

We use a simple homebrew Gridworld implementation. Each Gridworld position maps to a colour, reward, and terminal state indicator. There are two special colours for the world border and obstacles. Each time-step, an agent gets the colour index of the current agent's position. An agent is able to see obstacles or border colour when trying to go through them, which just replace current position colour for one step without actually changing the agent's position. We also punish an agent trying to go through a wall or world border in addition to base step reward punishment to facilitate faster convergence to optimal trajectories.

Samples of setups used in our experiments are depicted in Figure 6. Additional setups for scaling experiments are shown in Figure 7. Here, the red colour corresponds to obstacles (negative numbers) and the light gray colour to the reward (number 4). Letters A and G correspond to the agent's initial position and goal, correspondingly. Floor colours are resampled for each trial, but we ensure that there are no more than five floor colours, including the terminal state. In each episode, the agent and the terminal rewarding state have the same initial positions. The episode ends when the terminal state is entered or the limit of action steps is reached.

### F.2 ANIMALAI

AnimalAI is a testbed inspired by experiments with animals (Crosby et al., 2020). The environment consists of 3D area surrounded by a wall and many different objects that can be placed using a configuration file, including walls, food, ramps, trees, movable obstacles, and more. We tested our agent in a 10x10-meter room surrounded by walls (see Fig. 8). Each timestep, the agent gets an RGB image of a first-person view of the environment and is able to influence its behaviour by choosing one of three actions: turn left, turn right, or go forward. The agent and food item have the same fixed initial positions for each episode. The episode ends if the food item is collected or the maximum episode time is reached. To make an agent's actions more discretized, we repeat its choice three times and skip three frames.

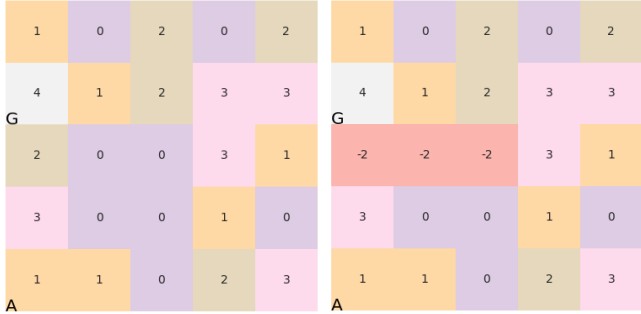

Figure 6: Schematic view of POMDP Gridworld setups. The left picture shows the initial setup of the environment, with the reward delivered by the light gray square (number 4). The right picture represents the environment after the reward is blocked by obstacles (red squares or negative numbers). The letter A depicts the agent's initial position and the letter G—the goal.

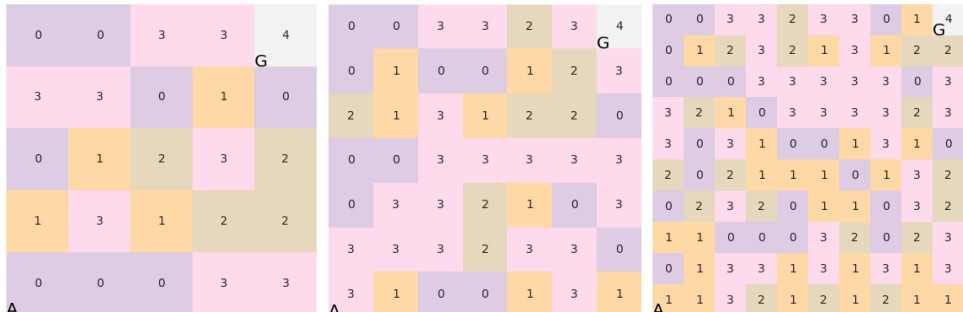

Figure 7: Schematic view of POMDP Gridworld setups for scale experiments. From left to right: 5x5, 7x7, 10x10. The letter A depicts the agent's initial position and the letter G—the goal.

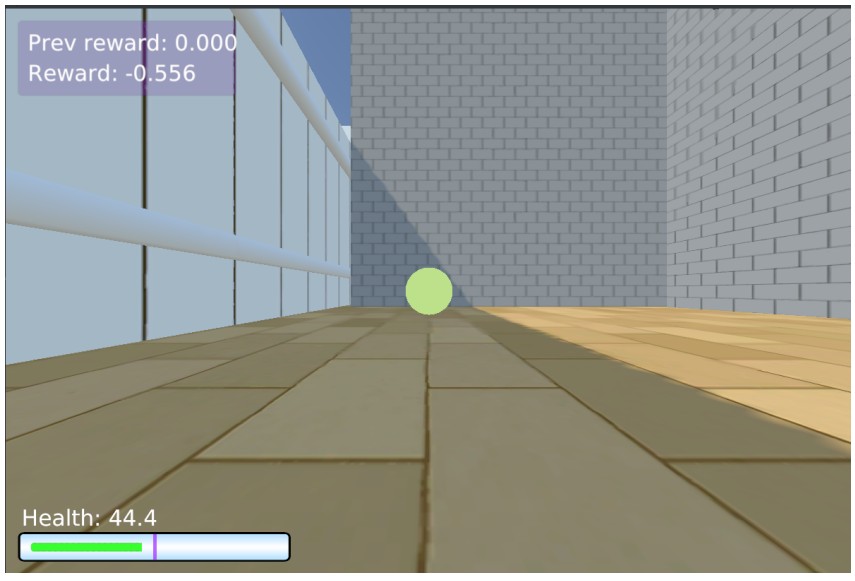

Figure 8: Sample of the agent's first-person view of our setup in AnimalAI environment. The green ball represents a goal, a food item to be collected.

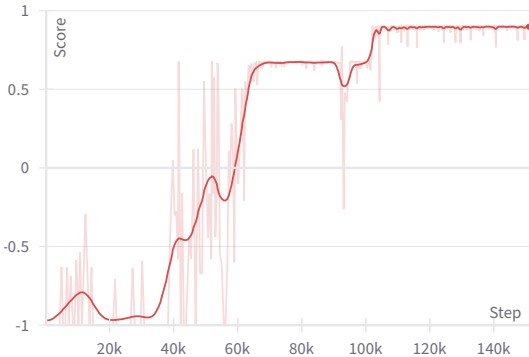

Figure 9: Dreamer V3 (18M) score in AnimalAI experiment described in this paper.

## G  CAPACITY

In this section, we provide basic theoretical analysis of DHTM's capacity. Since the hidden state for unpredicted feature state transition is generated randomly, the main source of errors is accidental collision of hidden states. Due to special form of the emission factor (see equation 6), the collision is possible only for the same feature sate. Here we analise the capacity of DHTM for factor grpah depicted in Figure 5. Let's assume that the stream of feature states is sampled from uniformly distributed categorical random variable. Therefore, the probability to have a collision $p_c = \frac{1}{n_\varphi n_v n_h}$, where $n_v$—number of hidden variables, $n_\varphi, n_h$ are the number of states for feature and hidden variables correspondingly. Therefore, the probability of having no collisions for a $l$ transitions is equal to $(1 - p_c)^l$. Let's set a threshold $c \in (0, 1)$, which determines our confidence in that the sequence of transitions is collision free. Solving the equation $1 - c = (1 - p_c)^l$ with respect to $l$, we get:

$$l = \frac{\log(1 - c)}{\log(1 - \frac{1}{n_\varphi n_v n_h})}. \tag{18}$$

Using the Taylor expansion for $n_\varphi n_v n_h \to +\inf$, we can see that $l$ grows linearly with the total number of states $n_\varphi n_v n_h$. Therefore, DHTM's capacity scales linearly with the number of neurons. This, however, can be easily improved if we set additional prior on hidden state activation probability, which would take into account hidden state usage, but we leave it for future research.

## H  ADDITIONAL EXPERIMENTS

### H.1  DREAMER

In order to ground our experiments to state-of-the-art model based algorithms, as an additional baseline, we test the smallest version of Dreamer V3 with 18 million parameters. We use the implementation described in Voudouris et al. (2023), which is based on Hafner et al. (2023) and adapted for AnimalAI.

We tested it with default parameters in the same setup as described in Section 4.2, except we don't apply frame skip. The agent successfully solves the task after $105 \times 10^3$ action steps. Taking into account the frame skip used in our experiments with DHTM, it's approximately $35 \times 10^3$ action steps versus $3.5 \times 10^3$ for the DHTM agent to solve the task.

### H.2  SCALABILITY

In this section, we explore DHTM's scalability properties and compare them to the EC agent. In RL experiments, like presented in section 4, the number of segments largely depends on the RL algorithm used, its exploration strategy, and how fast it converges to a stable policy. We tested DHTM and EC agents in empty Gridworld of different sizes: 5x5, 7x7 and 10x10, where the agent and the goal are placed in opposite corners (see Fig. 7). As shown in Figure 10A, the DHTM-based

agent requires fewer episodes to find the goal; however, the policy becomes less stable as the size of the Gridworld grows. As a consequence of faster convergence, DHTM produces fewer segments than the number of EC's dictionary entries throughout experiments. It also can be seen that for both DHTM and EC agents, the number of memory units grows non-linearly with the state space size (see Fig. 10B).

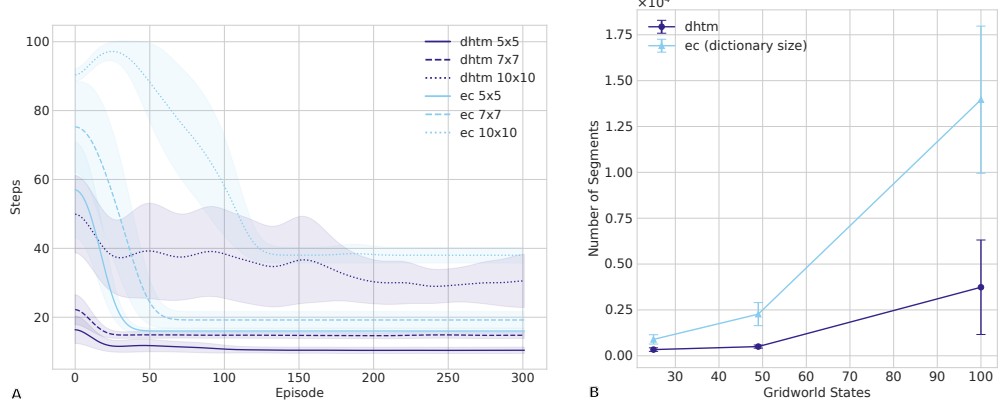

Figure 10: Comparison of EC and DHTM-based agents in foraging task in POMDP Gridworld of different sizes: 5x5, 7x7 and 10x10. **A.** Dynamic of steps required for the agent to reach the goal. **B.** Number of segments grown (or dictionary entries for EC agent) after 300 episodes with respect to state space size.

In principle, the number of segments reflects the number of transitions stored, which is proportional to the amount of experience provided to the memory. To show that, we tested DHTM and EC agent in 10x10 empty Gridworld using the same memory parameters as for experiments in section 4.1, but under uniform strategy. That is, agents freely explored the environment during 100 episodes and at the end the number of segments for DHTM and dictionary size for EC was recorded for each episode length. Figure 11 shows that DHTM's number of segments and EC's dictionary size grow linearly with the maximum episode length. In other words, since DHTM lacks generalisation abilities it largely serves as a trajectory buffer, in the same sense as a dictionary in EC.

DHTM's computational complexity (time and memory) of the prediction step grows linearly with respect to the number of segments. Therefore, to use it for huge environments, there is need to prune unnecessary segments or remember only selected episodes. This is a common problem among episodic-like memories. The current approach to form SF also should be adjusted to be scalable, since it requires maximum planning horizon to be equal to the episode length. Among the improved approaches could be to employ n-step TD learning, hierarchical planning or SF caching, to recalculate it less frequently.

## H.3    NOISE TOLERANCE

In order to show the key difference between DHTM and its simplified EC version, we tested agents in the same AnimalAI setup as described in 4.2, but we add one noise variable to the VAE encoder. It's easy to see that the naive mapping from distributed VAE encoding to one categorical variable used for EC doesn't work in this case, since even identical observations will likely be encoded differently due to the noise variable. In contrast, since DHTM is able to handle distributed representations directly (as the name implies), it still can use non-noise variables independently to learn transitions.

In this experiment, we vary DHTM's factor size (see Fig. 12). Since factors formed completely random, probability to include the noisy variable increases with factor size and this consequently worsen the performance of the agent. If the factor size is set to the total number of variables, DHTM becomes effectively unfactorised, similarly to its EC counterpart.

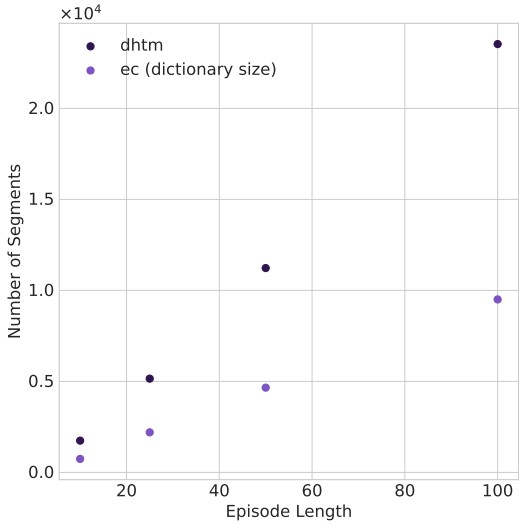

Figure 11: Comparison of DHTM's number of segments with EC agent's dictionary size depending on episode length in Gridworld after uniform exploration during 100 episodes.

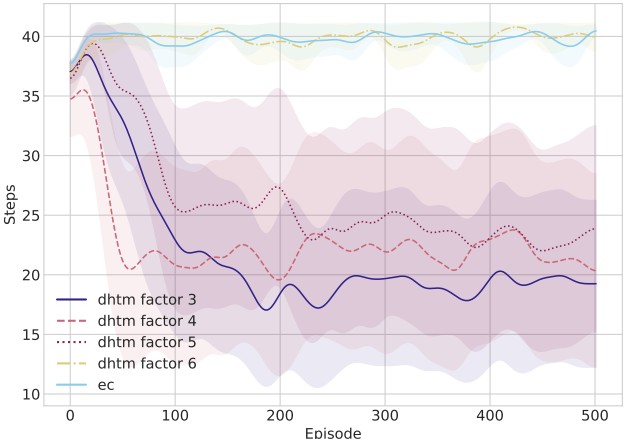

Figure 12: Number of action steps the agent takes to reach the goal in a 10x10-meter room in the AnimalAI environment, using DHTM and EC models with a distributed Categorical VAE encoder that includes one additional noisy variable (six variables in total). DHTM's factor size is varied, determining the number of variables sampled for each factor.

## I  GLOSSARY

**Categorical Random Variable**—a discrete random variable that can take on of finite $K$ possible states.

**Cortical Column or Minicolumn**—a population of neurons in the neocortex that spans across layers and shares sensory input.

**Dendritic segment**—a group of synapses (neuron's connections) that acts as an independent computational unit affecting the resulting neuron's activity.

**Factor Graph**—bipartite graph representing the factorization of a probability distribution, with one part representing factor nodes and another—random variables.

**Factor size**—number of variables connected to the factor. For simplicity, we refer only to the number of previous hidden states, used for prediction.

**Hidden Markov Model (HMM)**—statistical model of a stochastic process where state probability depends only on previous state of the process.

**Multi-compartment neuron model**—a model of neuron that divides neuron's connections into groups (segments) of different types (compartments), where each group may be considered as partly independent computational unit and groups of each compartment may affect the neuron's activity differently.

**Oracle**—a mythical creature that gives up to our algorithms insightful information about the environment that is usually hidden.

**Successor Representations (SR)**—a discounted sum of future [one-hot encoded] observations.

**Successor Features (SF)**—a generalization of SR, a discounted sum of future latent states.

**Temporal Memory (TM)**—in this work by this term we mean "memory for sequences".

