# OpenReview forum: "Learning Successor Features with Distributed Hebbian Temporal Memory"
_ICLR.cc/2025/Conference — ICLR 2025 Poster_

### Official Review · Reviewer_cQzX · 2024-10-29

**Soundness:** 3
**Presentation:** 3
**Contribution:** 2
**Rating:** 5
**Confidence:** 2

**Summary:**

This paper proposes a new bio-inspired algorithm, Distributed Hebbian Temporal Memory (DHTM), for online learning to make decisions under uncertainty. This algorithm resembles a model of episodic memory, or a dynamic trajectory buffer, which remembers and replays sequences of actions which were rewarding in the past. It is implemented by compartmentalized neurons, with dendrites that recognize particular previous states and encourage recall of associated actions, and learning is realized by a Hebbian-like rule on these dendrites. The method is evaluated on two navigation tasks (GridWorld and AnimalAI) and compared against a few baselines -- LSTM, CSCG, and a simplified episodic controller which it resembles in performance.

**Strengths:**

* Exploring how bio-inspired neural networks can implement and perform inference on statistical models is an interesting opportunity for interplay between AI and neuroscience.
* The experimental evaluation of the method is reasonable and with some more detail would support its effectiveness.
* The discussion of limitations of the method in the conclusion is thoughtful and thorough.

**Weaknesses:**

* The method is somewhat misrepresented in the introduction. What exactly it's meant to accomplish is not clear, and it's compared alongside very general learning methods which are meant to generalize effectively. After reading the entire paper, its focus seems to be on learning simple, non-generalizable strategies quickly.
* The description of the model is difficult to follow. A huge amount of notation is introduced, and much of it is not defined precisely and has to be inferred from context. This is most problematic in section 3.2, where the bio-inspired implementation is described. Ideally the neuron model would be laid out, and then the variables of the neuron model would be associated with variables from DHTM.
* Key details of the experiments, particularly how the baseline models were tuned, are missing and make it difficult to evaluate them.
* The paper presents a simple episodic control-based model which is highly competitive with the much more complicated DHTM which it focuses on. It is unclear what the advantages of DHTM are compared to this simple model.
* There is no substantial discussion of related work.

**Questions:**

* To my knowledge the term "temporal memory" is not in common use and should be briefly defined when it is used in the abstract.
* The distinction between requiring "complete data sequences to be available during training" (lines 53-54) and "access to only one sequence data point at a time" (lines 61-62) is not clear to me.
* "Factor graph formalism" should be defined in the introduction.
* "local Hebbian-like learning rules ... make the learning process much faster than gradient methods" (lines 70-71) should be revised to clarify in which sense "faster" is meant.
* Eq. (3) is unclear because of the summation over $\Omega_k$. I believe this is meant to express summation over all possible values of the "previous time step RV indexes [sic] included in the $F_c^k$ factor" which should be clarified.
* What is meant by "$f_l$ is proportional to several coincidences $s_l = c_t^j = 1$ in the recent past" (line )?
* Isn't the VAE encoder in section 4.2 significantly more expressive than the k-means encoder? Doesn't this give DHTM-vae an advantage as the only model with this representation?
* It is claimed that the paper shows that "unlike CSCG and table EC, DHTM is able to handle distributed representations and account for statistical dependencies between variables of these representations". I'm not sure where this was shown or how.
* It may be beneficial to revise the storytelling in the introduction to emphasize the points raised in the conclusion that this is a weak but very fast form of learning which may complement a slower, more accurate method over the course of learning.

---

> ### Author Response · Authors · 2024-11-21
>
> Thank you for your detailed review and valuable comments! We updated the manuscript taking into account your suggestions. The most important changes are highlighted and tagged for your convenience.
>
> ## W1 Misleading introduction
> Thank you for pointing this out! We revised the introduction, adding more emphasis on the type of the memory we present in this work.
>
> ## W2 DHTM’s description
> We substantially revised section 3.2 to make it easier to follow. We also moved most of the derivations to Appendix B, since they are too heavy for the first read.
>
> ## W3 Implementation details
> Due to paper size limits, we had to put most of the implementation details into Appendix E.
>
> ## W4 EC vs DHTM
> The EC version of the algorithm lacks some important features such as the ability to process distributed encoding, factoring hidden state space as well as probabilistic estimates, which might be crucial in stochastic environments. It’s also possible to modify the EC version, to add these features, however, it seems like the resulting algorithm would be very similar to full DHTM. At the current stage of DHTM’s development, there is, probably, no decisive experiment that would show DHTM’s advantage over EC. However, newly added experiments in Appendix H.2 might give an idea that, at this point, DHTM already has better scaling capacity and faster convergence.
>
> ## W5 Related work
> We added more on related work in the introduction.
>
> ## Q1 Temporal Memory
> We agree that the term “Temporal Memory” might be confusing in the abstract, so we removed it.
>
> ## Q2 Complete data sequences
> There is a common challenge in continual learning that algorithms should be able to incorporate new sequential data into the system by small fragments, and without access to previously seen data (Khetarpal et al. 2022). Training only on small amounts of new data (at the extreme, just one sequence element) usually results in overfitting for deep learning methods and loss of previously acquired knowledge, known as catastrophic forgetting. Therefore, contemporary deep learning methods rely on huge experience buffers that are sampled to calculate parameter updates. That’s what we understand by “complete data sequences”.
>
> ## Q3 Factor graph formalism
> We added more information about factor graphs in the introduction and background sections.
>
> ## Q4 Fast Hebbian learning
> Here “faster” means that algorithms relying on Hebbian learning are usually more sample efficient, and require less training steps to converge in comparison to stochastic gradient descent (see for example Journé et al. 2023).
>
> ## Q5 “Eq. (3) is unclear”
> We added corresponding clarification.
>
> ## Q6 Meaning of $f_l$
> It simply means that $f_l$ is an exponential average of the segment's activation frequency, given that its root cell is also active.
>
> ## Q7, Q8 VAE encoder with other models
> We updated AnimalAI experiments by adding other models with VAE encoder for comparison. Since EC and CSCG have only one categorical variable as an input, each unique VAE representation should be mapped to the corresponding state of one categorical variable. For Categorical VAE with $5$ categorical variables with $50$ states each, it means that the resulting input variable should have $50^5$ states to have the same expressive power as distributed encoding. However, in practice, the total number of unique representations is much less for a particular environment and task. For example, in AnimalAI, it’s only about $1200$ VAE patterns. Still, CSCG with such a large state space would be infeasible to train. But since EC doesn’t require allocating all states in advance, it’s possible to train EC baseline trained on VAE representations, which, however, can’t use possible factorization in VAE encoding space in contrast to DHTM.
>
> ***
> Khetarpal, Khimya, Matthew Riemer, Irina Rish, and Doina Precup. 2022. ‘Towards Continual Reinforcement Learning: A Review and Perspectives’. Journal of Artificial Intelligence Research 75 (December):1401–76. https://doi.org/10.1613/jair.1.13673.
>
> Journé, Adrien, Hector Garcia Rodriguez, Qinghai Guo, and Timoleon Moraitis. 2023. ‘Hebbian Deep Learning Without Feedback’. arXiv. https://doi.org/10.48550/arXiv.2209.11883.

---

> > ### Comment · Reviewer_cQzX · 2024-11-21
> >
> > Thank you for the thoughtful response -- my comments regarding the presentation have largely been addressed, and I'm happy to increase soundness and presentation to 3 and my overall score to 5. I appreciate your candor in the comparison of DHTM to EC, but I think I would need to see an experiment that convincingly separates the performance of the two to substantially increase my estimation of the contribution.

---

> > > ### Author Response · Authors · 2024-11-24
> > > **EC vs DHTM**
> > >
> > > Thank you for your fast response! As noted before, distributed implementation, which allows for hidden space factorization, is a key feature of DHTM. EC, on the other hand, represents a simplified counterpart without this feature. To exemplify its importance, we conducted an additional experiment in which we added one noise variable to categorical VAE variables in the AnimalAI foraging task (please have a look at Appendix H.3 “Noise Tolerance”). The result shows that, in contrast to EC, DHTM with proper factorization still makes useful predictions, steering the agent to the goal.

---

### Official Review · Reviewer_UWu5 · 2024-10-31

**Soundness:** 3
**Presentation:** 3
**Contribution:** 3
**Rating:** 6
**Confidence:** 2

**Summary:**

The paper introduces Distributed Hebbian Temporal Memory (DHTM), a model inspired by the HTM neuron (modeled after cortical columns) for spatial-temporal learning. The model takes in a sequence of observations generated from a trajectory through an environment and generates a hidden state representation. These are then used to predict the cumulative distribution of future states under a uniform policy to form a Successor Feature (SF) representation. This is then subsequently used by an RL agent to estimate the Q value (assuming the reward can be decomposed linearly) for action selection.

The authors perform experiments on GridWorld and AnimalAI environments and compare DHTM to LSTM and CSCG baselines, showing that it outperforms both, reaching the goal state in a few number of states in fewer episodes.

**Strengths:**

- The model proposed by the authors is novel and formulated using the factor graph framework, making it theoretically-grounded, allowing for techniques in that area to be easily applied.
- The model is online and uses local learning rules, which makes it more applicable.
- The model is biologically inspired and the authors make specific connections to its neural implementation/plausibility.
- The authors provide an interpretation of the model in terms of episodic control.

**Weaknesses:**

- The description of DHTM is difficult to follow. I suggest adding paragraph-level divisions to clarify the logical structure of the sections, or at least outline the general structure of the section at the beginning.
- Since the model uses a factor graph formalism, I think having a section for that in the background would improve clarity.

**Questions:**

- How does the DHTM scale? The experiments applied the model to 5x5 and 10x10 environments. What if the environment had a larger number of states and a larger number of actions? How does this affect the computational complexity and loss curve?
- How well does DHTM handle partial observability? Can it distinguish aliased observations well?
- CSCG has been shown to learn the spatial structure of the environment which allows for generalization during navigation. Can DHTM do the same?

---

> ### Author Response · Authors · 2024-11-21
>
> Thank you for your thoughtful questions and commentaries! We uploaded a revised version of our paper that, hopefully, addresses your concerns and takes into account your suggestions.
>
> ## W1 Model description
> We substantially revised DHTM’s description, simplifying it and moving the most difficult part into Appendix B.
>
> ## W2 Background
> We added more information about Factor graphs into the background section about HMM.
>
> ## Q1 Scalability
> We made additional experiments to investigate DHTM’s scaling properties and compare it to EC agent in Appendix H.2.
>
> ## Q2 Aliased observations
> DHTM was specifically designed to disentangle aliased observations, since it's a common problem in POMDP environments. DHTM is a high-order memory that is able to assign different hidden states to identical observations. We show it in our POMDP Gridworld navigation experiments, where many positions have identical colours. Only memory that is able to handle aliased observations will successfully solve the task.
>
> ## Q3 Generalization
> We revised our introduction section to stress more clearly that, in contrast to CSCG, DHTM is closer to episodic-like memory, and it is not able to generalize experience forming schemes. By comparison to CSCG, we aimed to show that, in certain tasks, it’s more advantageous to use fast episodic types of memories than slowly learning semantic memories like CSCG.

---

> > ### Comment · Reviewer_UWu5 · 2024-11-26
> >
> > Thank you for the reply. I believe the changes warrant an increase in presentation score. However, I would like to see experiments in more complex/larger-scale environments before considering a change in the overall rating.

---

### Official Review · Reviewer_uBbj · 2024-11-05

**Soundness:** 3
**Presentation:** 3
**Contribution:** 3
**Rating:** 6
**Confidence:** 3

**Summary:**

This paper introduces a novel algorithm for online temporal memory learning in decision-making tasks with non-stationary / partially-observable environments. Distributed Hebbian Temporal Memory is a biologically-inspired algorithm that uses Hebbian learning rules in order to learn Successor Features, a powerful framework in reinforcement learning to decouple the environment's dynamics from the reward function.

**Strengths:**

- DHTM operates fully online using local learning rules due to its local Hebbian learning rules, making it well-suited for non-stationary environments without needing explicit resets or retraining. Its ability to dynamically form and update memory representations allows it to cope with sudden changes in the environment effectively. Furthermore, you can train it fully online since it doesn't require backpropagation through time.
- The algorithm is computationally efficient due to sparse transition matrices and distributed representations, and could be potentially implemented efficiently in neuromorphic hardware.
- The experiments indicate that DHTM could achieve higher sample efficiency compared to some state-of-the-art models. For example, in the AnimalAI environment, DHTM requires significantly fewer interactions with the environment to learn effective policies than models like Dreamer V3.

**Weaknesses:**

- The unique encoding of each observation sequence limits the model's ability to generalize across different sequences which represent the same underlying state.
- The performance of the algorithm is very close to that of Episodic Control on some tasks, which leads to a question of its advantage over a similar method.
- The baseline models like LSTM may not be fully optimized for online learning, potentially skewing the comparison results.
- The paper lacks a thorough theoretical analysis of the algorithm's properties such as convergence guarantees and computational complexity.
- I think you should have more exposition explaining what CSCG, Episodic Control, and Dreamer V3 are.
- I am willing to improve my score if you address some of the questions posed in the following section.

**Questions:**

- Can you talk about how to incorporate mechanisms that enable the model to generalize across different sequences leading to the same state. Would hierarchical or abstract representations be useful here?
- It would be nice to see a experiment that shows a substantial improvement of the DHTM model over Episodic Control, or an explanation of why it's better to implement the full algorithm.
- In the AnimalAI task, you only test DHTM with VAE whereas all of the other algorithms are only tests with k-means. Is there a reason why you can only test DHTM with VAE and not the others? If it's possible to test the others with VAE, I would be interested to see that in a comparison.
- It would be worthwhile to see the performance of Modern RNN architectures (such as LRU or RKWV mentioned in the beginning of the paper) instead of LSTM, unless there is a reason those architectures don't apply to these tasks. These algorithms also avoid using BPTT and are quite performant.
- Can you comment on how the architecture compares to other Temporal Memory architectures that are based on Hebbian Learning Rules such as Sequential Hopfield Networks? For example, how does it relate to Chaudhry et al. 2024 (Long Sequence Hopfield Memory) or Tang et al. 2024 (Sequential memory with temporal predictive coding). I think you should add a more thorough literature review of temporal memory.
- I would like to see the effect of an oracle on Episodic Control and DHTM in Figure 4.

---

> ### Author Response · Authors · 2024-11-21
>
> Thank you for your questions and valuable comments! We did our best to address your concerns and highlighted important changes in the new revision for your convenience.
>
> ## Q1 Generalization
> We think that hierarchical approach is the most promising for online generalization in DHTM. We see current DHTM implementation as a first level in the hierarchy, which aims to remember as much experience as possible. We are in the process of developing the second level, which shares some properties with DHTM, but its main purpose is to connect hidden states from the first level into clusters according to their n-step predictions. Furthermore, we aim to make the second level non-destructive, so that in case of generalization error we don’t lose potentially valuable experience on the first level.
>
> ## Q2 DHTM vs EC
> The EC version of the algorithm lacks some important features such as the ability to process distributed encoding, factoring hidden state space as well as probabilistic estimates, which might be crucial in stochastic environments. It’s also possible to modify the EC version, to add these features, however, it seems like the resulting algorithm would be very similar to full DHTM. At the current stage of DHTM’s development, there is, probably, no decisive experiment that would show DHTM’s advantage over EC. However, newly added experiments in Appendix H.2 might give an idea that, at this point, DHTM already has better scaling capacity and faster convergence.
>
> ## Q3 VAE encoder with other models
> We updated AnimalAI experiments by adding other models with VAE encoder for comparison. Since EC and CSCG have only one categorical variable as an input, each unique VAE representation should be mapped to the corresponding state of one categorical variable. For Categorical VAE with $5$ categorical variables with $50$ states each, it means that the resulting input variable should have $50^5$ states to have the same expressive power as distributed encoding. However, in practice, the total number of unique representations is much less for a particular environment and task. For example, in AnimalAI, it’s only about $1200$ VAE patterns. Still, CSCG with such a large state space would be infeasible to train. But since EC doesn’t require allocating all states in advance, it’s possible to train EC baseline trained on VAE representations, which, however, can’t use possible factorization in VAE encoding space in contrast to DHTM.
>
> ## Q4 Adding new baselines
> We updated the experiments in the new revision of the manuscript by adding also RWKV for comparison. We found that, in our setup, there is not much performance difference between RWKV and LSTM, except that RWKV of the same hidden state size as LSTM is a bit slower to train. Though their losses are decreasing with training, the performance stays approximately the same in AnimalAI. Apparently, LSTM and RWKV are not apt for probabilistic multistep predictions and, perhaps, require more architectural adjustments that are beyond the scope of this work.
>
> ## Q5 Related papers
> Thank you for pointing this out! We added a corresponding paragraph to the introduction section.
>
> ## Q6 Oracle and EC
> We updated Gridworld experiments by adding EC with Oracle. Interestingly, EC with complete memory clean-up converges a bit slower. Apparently, it loses some experience that it could reuse otherwise.

---

> ### Author Response · Authors · 2024-11-27
> **Additional experiments**
>
> We are pleased to inform you that we have updated the manuscript by adding another experiment addressing your question about the difference between EC and DHTM (see Appendix H.3 "Noise Tolerance"). If you have any further questions, please let us know.

---

> > ### Comment · Reviewer_uBbj · 2024-12-02
> > **Increase in Score**
> >
> > Thank you for the updated experiments and clarification! I think it definitely warrants an increase in score. I especially liked the new experiments you did showing the advantage of DHTM over EC in Appendix H.3, I think it's worthwhile to highlight that in your final manuscript to make it clear what the benefit of this additional architecture is. I also think that the second level you mentioned in your generalization seems quite important for the development / value of this entire algorithm, so while I will increase the score I also suggest that it might be worthwhile to wait until you've finished with that portion and publish it together for more substantial impact.

---

> > > ### Author Response · Authors · 2024-12-03
> > >
> > > Thank you for your appreciation of our work and your advice! Though the second level should definitely add value to our work, we showed that DTHM can be used as a standalone algorithm for episodic control. We also noted in conclusion that DHTM can be potentially used as a fast adaptive buffer (i.e., the first level) for any slow generalizing memory (the second level). In the future work, we plan to test different generalizing algorithms as the second level interchangeably, including the one we are developing now. Therefore, we believe that DHTM could be presented separately so that interested researchers could test other second-level algorithms concurrently.

---

### Official Review · Reviewer_JqYM · 2024-11-07

**Soundness:** 3
**Presentation:** 2
**Contribution:** 4
**Rating:** 6
**Confidence:** 4

**Summary:**

This paper proposes a novel algorithm for learning successor features for reinforcement learning agents in non-stationary partially observable environments. The main feature of the algorithm is a new episodic memory architecture that combines ideas from Factor graphs, Hidden Markov Models and episodic control, along with a neuron concept similar to the Hierarchical Temporal Memory, which is in itself inspired by biological neurons as complex units with spatio-temporal computation. Features of this new type of memory include, efficiency of computation, distributed representations and a local (Hebbian) learning rule. The algorithm is embedded in a simple RL agent and tested, both, in a simple gridworld environment and in a more structure AnimalRL environment.

**Strengths:**

This paper is an very interesting contribution. The connection of successor features, factor graphs, episodic control and complex neuron morphologies is very attractive. In particular, it offers a practical computational interpretation of the role of complex morphologies in biological neural network: computing factors/features based on previous experiences. The paper covers a lot of theoretical ground in connecting the different ideas used to build the model. In general each aspect of the model is theoretically sound and the experiments are tested with a good set of baselines. I will try to be comprehensive in the weaknesses and questions with the intentions of making the contribution stronger.

**Weaknesses:**

1. The basis of comparison with the other baselines in experiment number 1, is the capacity of the model to forget, however, in algorithm 1, the DHTM agent includes a memory_reset() step. This seems to invalidate the whole argument of this experiment.
2. The connection with biology is only superficial, but I think it can be made stronger. It is not clear if some of the things I am going to mention are already in the authors minds, but they are definitely not clear in the text:
- The activity of each of the neurons in the model is associated with a spike only makes sense if there are two different timescales at play, so that you can collect many spikes before making a decision (this is what firing rate means). Do the neurons only spike once per episode/state visited?
- The appendix does not explain very clearly the connection of your model with HTM is not very well explained (and the citation of Hawkins paper is not the appropriate paper, choose one in which they explain the model). For example, it is never clear what you mean by the receptive field of a cell.
- Explain better how the computation of Fe relates to the columnar structure of the cortex? The authors seem to be confusing the morphology of one neuron with the structure of the cortex (this is related to the connection with HTM above). One is related with the sort of computations performed, the other with the origin and information carried by the afferents.

3. The presentation style is probably the weakest point of this paper. It is very difficult to follow. At the start, the definition of a POMDP is very difficult to read, adding a bit more words can make this and other definitions easier to read (if they are in the paper, they will be read by someone, otherwise they can go in the appendix).
- Many things are explained in the wrong place, before or after they appear in any equation. For example rec(l) for the receptive field, or the definition of log Li.
- The calculation of the emission factors E is never shown explicitly! but it appears in equation 7
- Equation 7 is not well justify at all, it is very difficult to see how you go from (4) to (7)
- Many times it is not clear what is part of your algorithm and what is you explaining the old way of doing things, for example: line 191- "the feature variables are considered independent (in this work?), however, the are interdependent (so which?).
- Similar to the previous one, in the exposition of algorithm 2, it seems that you are sometimes talking about DHTM and sometimes about the naive EC agent.

4. The scope of the paper seems a bit misleading. If the focus is the learning of successor representations for non-stationary environments, why not testing it in more standard and more complicated environments than the ones shown in this paper? The experiments are barely non-stationary and not the usual testing ground for comparison with the baselines chosen. As I said, I think this is a very interesting idea but I think the scope needs to be made more realistic or add the necessary tests. This makes me think that there is some problem with scaling the presented ideas; if that is the case what is it?
- Do the number of segments grow fast or exponentially with the complexity of the environment (please show evidence otherwise). The factors are defined in terms of combinations of ALL the RVs involved!
- Is the computation as efficient as suggested? Is having segments more efficient as claimed? How fast? (the results are shown in action steps but no mention about memory/time efficiency)
- What is the capacity of the memory?

5. Definitely needs more figures! figure 1 is infinitely complex, it has a lot of information (more than 3 sections worth of information). This figure can be split to illustrate different aspects of the exposition. For example, the details of the computation of LogLi can be added in a later figure. In relation to the figures, figure 6 also needs more information. Mark what the goal and the start position is (the colors have barely any contrast and someone with visual impairments could not get any information from this figure)

6. Others
- It is not clear to me how are new segments created
- the change from Fe to fl for the emission factors was confusing, please add some warning
- The calculation of the conditional probability on the features in 294 is only tangentially mentioned, but it is very important, here is the only time the features are introduced, and it deserves some discussion
- How is the planning horizon "T" determined. How are you measuring the shape of the distributions to determine this parameter?
- The explanation of the Episodic control agent needs to be rewritten for clarity.
- In equation 10, make clear that Rt is the reward coming from the environment (there are other definitions for Rt before)
- It took me some time to understand that each neuron is a state, and each neuron is also a factor. It is not clear how the population of neurons is set up. It seems that we would need to keep adding neurons with longer experiences? Again, it will be very helpful if the architecture and computations are separated from figure 1.
- 291,292, please explain this step.

**Questions:**

Any of the weakness are also questions.

---

> ### Author Response · Authors · 2024-11-21
>
> Thank you for such a detailed review! We updated the manuscript and hope that we managed to improve its quality by incorporating your suggestions and addressing your questions. We added highlights of important changes to the revised text, and you can use text search by your id to find them.
>
> ## W1 RESET_MEMORY
> Though, indeed, the name of the procedure might seem counterintuitive, we elaborate in section 3.4 that `RESET_MEMORY` procedure concerns only memory state or its current activity, but not its parameters (synaptic weights), which are usually considered the main carriers of the experience in the models we tested.
>
> ## W2 Biological interpretation
> **About spikes:** Thank you for noting that! Indeed, we have only one timescale, so we agree that calling cells activity as spikes might be confusing here, therefore we removed it.
>
> **Connection to HTM:** Certainly, you are right, we’ve changed the reference to a more suitable paper. We also added clarification on DHTM’s connection to HTM in Appendix C.
>
> **Columnar structure:** This form of the emission factor means that one feature state (input cell) influences several hidden states (cells in column) and only them. That is, several hidden states share input. We use the same emission factor that CSCG uses, which also refers to columnar structure of the neocortex, and it’s in concordance with HTM’s implementation of columnar structure.
>
> ## W3 Presentation style
> Thank you for your suggestions! We substantially revised Background and Neural Implementation sections in order to simplify them. We agree that the details of the derivation might be confusing and are difficult to comprehend from the first read, so we decided to move most of the details to Appendix B.
>
> ## W4 Scope of the paper
> _“The experiments are barely non-stationary”_ - there are different types of non-stationarity, here we refer to piecewise non-stationarity of the environment’s transition function according to taxonomy used in Khetarpal et al. 2022. Though the task might appear trivial, it’s still challenging for most of the contemporary deep learning methods, which we aimed to show in our research.
>
> Our method indeed has some scaling challenges, which, however, are common for episodic-like memories. We added additional experiments in Appendix H.2 and discussed DHTM’s scaling properties there. We also provided theoretical analysis of DHTM’s capacity In Appendix G.
>
> **About segment efficiency**: We meant that in contrast to conventional HMM, we use sparse representation of transition matrix in the form of segments. In that case, prediction step time complexity grows linearly with the amount of segments and with hidden state space size. In contrast, if a full transition matrix is used, the time complexity of the prediction step grows quadratically with the state space size. For example, CSCG with the same state space size as DHTM in our experiments would be infeasible to train.
>
> ## W5 Figures
> Thank you for your feedback!  We agree that Figure 1 could be improved and we will take it into account in future revisions of the manuscript. Good point, about Figure 6, we added annotations.
>
> ## W6 Others
> Thank you for your questions, we did our best to add more clarifications where it’s possible.
> However, there are some misunderstandings, which we should address directly:
>
> _“neuron is also a factor”_ - it’s not true, a factor is represented by a collection of segments, so we need to grow only segments to store new experiences. We hope that in the revised text it’s more clear. Populations of neurons are set up at the onset of learning and correspond to RV’s state space sizes.
>
> _“291,292, please explain this step”_ - we refer to the fact that if one of the terms in the sum under logarithm $\log(x_1+x_2+x_3+...)$ is much larger than others, then we can approximate this sum by taking a maximum instead. Indeed, let’s assume that $ x_m = \mathrm{max}(x_1, x_2, x_3, …) $, then $\log(x_1+x_2+x_3+...) =$$ \log x_m(1+x_1/x_m+x_2/x_m+x_3/x_m + ...) = $$\log x_m + \log (1+x_1/x_m+x_2/x_m+x_3/x_m + ...)$$ \approx \log \mathrm{max}(x_1, x_2, x_3, …)$, when $x_m >> x_1, x_2, x_3, ...$. Interestingly, that this operation is similar to logical `OR` which is used to aggregate segments' activity in the HTM neuron and also has been found in some biological neurons as noted in Stuart & Spruston, 2015.
> ***
> Khetarpal, Khimya, Matthew Riemer, Irina Rish, and Doina Precup. 2022. ‘Towards Continual Reinforcement Learning: A Review and Perspectives’. Journal of Artificial Intelligence Research 75 (December):1401–76. https://doi.org/10.1613/jair.1.13673.
>
> Stuart, Greg J., and Nelson Spruston. 2015. ‘Dendritic Integration: 60 Years of Progress’. Nature Neuroscience 18 (12): 1713–21. https://doi.org/10.1038/nn.4157.

---

### Author Response · Authors · 2024-11-29
**General Response**

We thank all the reviewers for their comments and questions, which helped us to improve the manuscript's quality. In the updated version of the work, we aimed to address all the paper’s weaknesses and reviewers’ questions:

* Reviewers noted that the scope of the paper is not clear from the introduction, and the methods section is difficult to follow. We revised the introduction by adding emphasis on the type of memory we aimed to implement and reviewing more related work. Background and method sections were simplified, and most of the heavy mathematical details were moved to the appendix.

* Reviewers needed more context to our experiments, so we added another baseline: contemporary RNN memory RWKV. As justly noted, some of the baselines could also employ VAE, so we added corresponding experiments.

* The major concern was about DHTM’s scalability, so we made additional experiments and put our related discussion into Appendix H.2. In short, DHTM’s computational complexity grows linearly with the amount of experience; therefore, it’s possible to test it in more challenging environments as a component of a more advanced agent’s architecture, which we leave for future work.

* Another common question relates to the advantage of DHTM before its simplified EC version. Firstly, experiments in Appendix H.2 show that DHTM is more sample efficient, perhaps due to better generalization properties. We also added an experiment exemplifying the importance of DHTM’s distributed implementation to Appendix H.3.

We hope that we successfully addressed all the concerns, and we are ready to answer any follow-up question.

The source code of the experiments was updated and is available at https://anonymous.4open.science/r/dhtm-FA0E

---

### Meta-Review · Area_Chair_s4hH · 2024-12-24

**Metareview:**

This paper introduces a novel, biologically-inspired algorithm (DHTM) designed for online sequence learning for decision making in non-stationary, partially observable environments. Leveraging factor graph formalism and Hebbian-like local learning rules, DHTM captures Successor Features (SFs) to facilitate decision-making without the need for backpropagation through time. The authors demonstrate the algorithm’s efficacy in tasks requiring episodic memory, such as GridWorld and AnimalAI, showing that it outperforms LSTM, RWKV, and CSCG baselines in terms of sample efficiency and adaptability. Three of the four reviewers voted for acceptance.

**Additional Comments On Reviewer Discussion:**

While the paper’s contributions are notable, reviewers raised concerns about clarity, scalability, and comparative advantages over simpler methods like Episodic Control (EC). The authors have significantly improved the manuscript through revisions, addressing reviewer feedback by adding baseline experiments, clarifying technical descriptions, and enhancing the discussion of scalability.

---

### Decision · Program_Chairs · 2025-01-22

Accept (Poster)